# Cu(I)- and Pd(0)-Catalyzed Arylation of Oxadiamines with Fluorinated Halogenobenzenes: Comparison of Efficiency

**DOI:** 10.3390/molecules25051084

**Published:** 2020-02-28

**Authors:** Maria S. Lyakhovich, Alexei D. Averin, Olga K. Grigorova, Vitaly A. Roznyatovsky, Olga A. Maloshitskaya, Irina P. Beletskaya

**Affiliations:** 1Department of Chemistry, Lomonosov Moscow State University, Leninskie Gory, 119991 Moscow, Russia; lyakhovich.chem@gmail.com (M.S.L.); grigorovao@gmail.com (O.K.G.); vit.rozn@nmr.chem.msu.ru (V.A.R.); maloshitskaya@mail.ru (O.A.M.); beletska@org.chem.msu.ru (I.P.B.); 2A. N. Frumkin Institute of Physical Chemistry and Electrochemistry, 31-4 Leninskii prosp., 119991 Moscow, Russia

**Keywords:** fluorinated aromatics, oxadiamines, Pd catalysis, Cu catalysis, amination

## Abstract

The comparison of the possibilities of Pd- and Cu-catalyzed amination reactions using fluorine-containing aryl bromides and iodides with oxadiamines to produce their *N,N′*-diaryl derivatives was carried out. The dependence of the reactivity of the aryl halides on the nature of the substituents and halogen atoms as well as on the structure of oxadiamines was investigated. It was found that the copper-catalyzed reactions were somewhat comparable with the palladium-mediated processes in the majority of cases, especially in the reactions with *para*-fluorine- and *para*-(trifluoromethyl)-substituted aryl halides, although the necessity to use aryl iodides in the Cu(I)-catalyzed amination was obvious. Pd catalysis was found inevitable for the successful amination of more sterically hindered *ortho*-(trifluoromethyl)aryl bromides.

## 1. Introduction

Fluorine-containing compounds represent a substantial part of modern medicaments. The majority of such compounds possess this element as a fluoro- or trifluoromethyl substituent in the aromatic ring [1]. The spectrum of their activities is impressive: they are used as non-steroid anti-inflammatory drugs, antibiotics, and anticancer agents; additionally, some of them possess anxiolytic, antipsychotic, and neurotropic activities [2,3,4]. Thus, great attention is paid to the elaboration of modern synthetic approaches to various fluorine-containing compounds [5,6]. Also of interest are bioactive fluoro- and trifluorosubstituted pyridines possessing aminoalkyl substituents [7,8]. This reinforces the importance of studies on convenient synthetic routes to these compounds, among which, the catalytic amination of corresponding halogenosubstituted fluorinated (hetero)aromatic compounds can be regarded as most straightforward and versatile procedure. Palladium-catalyzed amination of (hetero)aryl halides is well-known and widely described in the literature [9,10,11]. During last decade, an important trend in the catalytic organic synthesis has been clearly exhibited, that is, the so-called renaissance of Ullmann chemistry, which is extremely important for the development of new strategies of C–N bond formation [12,13,14,15,16,17]. Our own interest in this field deals with the catalytic arylation and heteroarylation of polyamines [18,19] and adamantane-containing amines and diamines [20,21]. We demonstrated the possibility of successful copper-catalyzed arylation of di-, tri-, and tetraamines as well as of adamantylated amines with fluoroiodobenzenes, (trifluoromethyl) iodobenzenes and fluorine-containing pyridines [18,22,23]. In the previous research, the most efficient catalytic systems for the copper-mediated arylation were found out.

In the present work, we focused on the arylation and heteroarylation of polyoxadiamines. These molecules are used as flexible linkers in biologically active compounds due to their hydrophilicity and chelating properties. For example, symmetrically disubstituted trioxadiamine A demonstrated antimicrobial activity [24], whereas its analog B was shown to possess antiprotozoal activity against *Trypanosoma brucei rhodesiense* (Figure 1) [25]. *N,N′*-bis(8-chloroquinolin-4-yl) derivatives of mono-, di-, and trioxadiamines were tested for antimalarial activity in [26], and the influence of the nature and length of the polyoxadiamine linker on the efficiency of the anticancer agents was studied in [27]. All compounds under investigation were symmetrically *N,N′*-disubstituted derivatives and it was found out that such double substitution may enhance the effect in comparison with a parent (hetero)aromatic compound without an oxadiamine linker.

Here, we report a thorough study of the *N,N′*-di(hetero)arylation of some oxadiamines using fluoro- and trifluoromethyl-substituted halogenobenzenes, which differ in the electronic and steric effects of the fluorine-containing substituents. A comparison of Cu(I)- and Pd(0)-catalyzed processes is carried out to demonstrate the scope and limitations of these alternative approaches.

## 2. Results and Discussion

### 2.1. Catalytic Amination of Halogenofluorobenzenes

Three oxadiamines (**1**–**3**) were chosen in the present research for the investigation of the catalytic arylation (Scheme 1) differing by the number of oxygen atoms and methylene groups between O and N atoms that may result in different binding with the catalytic species and, consequently, in different reactivity. The reactions were run in the presence of CuI/L1 (L1 = 2-isobutyrylcyclohexanone) or CuI/L2 (L2 = *rac*-2,2′-binaphthol (BINOL)) (method A). We employed 2.5 equivalents of iodobenzene in absolute DMF at 140 °C with 0.5 M concentration of oxadiamines. Cs_2_CO_3_ (2.5 equivalents) was used as a base and the reactions were conducted for 24 h to ensure the full consumption of the oxadiamines (Scheme 1). These reaction conditions were found by us earlier to be optimal for the diarylation of di- and polyamines [28]. The necessity to apply high catalyst loadings in the case of the CuI/L1 system (20/40 mol%) was shown to be important in our previous investigations of the trioxadiamine **1** diarylation with the simplest iodobenzene as well as in the arylation of adamantane-containing amines [29]. The composition of the reaction mixtures was analyzed by ^1^H NMR spectra, and the target compounds and by-products were isolated by the column chromatography on silica gel. The tables give preparative yields of the products if not otherwise stated.

The reaction of trioxadiamine **1** with 4-fluoroiodobenzene in the presence of CuI/L1 provided 24% yield of the target diarylated product **4** (Table 1, entry 1). The change of L1 for L2 ligand gave even a worse result (entry 2). The application of a less active 4-bromofluorobenzene increased the yield of **4** up to 35% (entry 3). The analysis of the reaction mixtures in all experiments revealed that other processes other than catalytic amination took place in all cases. The diarylation with 3-fluoroiodobenzene occurred smoothly (entry 4), and even with a bulkier *ortho*-isomer it gave 77% yield of the product **6** (entry 5). However, in the case of 2-bromofluorobenzene, the activity of the bromine was insufficient and general conversion of NH_2_ groups into NHAr did not surpass 40%, thus the chromatographic isolation was not undertaken. The selectivity in the case of 2,4-difluoroiodobenzene was low and the reaction products **13** and **13a** were isolated in low yields (20% and 19%, respectively).

In general, the reactions of dioxadiamine **2** with the same dihalogenobenzenes occurred in a similar manner. 4-Bromofluorobenzene provided a better yield of the diarylation product **7** than 4-fluoroiodobenzene (entries 6 and 8); however, the catalytic system with BINOL ligand proved to be more efficient (entry 7). The best yield of the diarylation product again was observed in the reaction with 3-fluoroiodobenzene (entry 9), and with a more sterically demanding *ortho*-isomer, monoarylated compound **9a** was isolated along with the main product **9** (entry 10).

Dioxadiamine **3** differs from the diamine **2** by the increase of the number of methylene groups between N and O atoms. It may result in the formation of less stable complexes with copper which will inevitably modify the reactivity in Cu(I)-mediated arylation. Thus, the reaction of this dioxadiamine with 4-fluoroiodobenzene led to 58% yield of the corresponding diarylated compound **10** (entry 11). This result is better than with oxadiamines **1** and **2**. On the other hand, the yield of the same product **10** in the reaction with 4-bromofluorobenzene was lower (entry 12). The arylation with 3-fluoroiodobenzene was successful (entry 13), but the more sterically hindered 2-fluoroiodobenzene provided a modest result (entry 14). Taking these facts into consideration, one may assume that the reactivity of the oxadiamine **3** is lower than that of **1** and **2**, and it helps to obtain better yields in the case of more active aryl iodides, while hampers the reactions with less reactive compounds. This correlates with the data obtained by us earlier with Cu(I)-catalyzed arylation of the polyamines [18]. Alternative Pd(0)-catalyzed arylation (Scheme 1, method B) of the oxadiamines **1**–**3** was studied using isomeric bromofluorobenzenes as they are preferable compared to iodoarenes.

Pd(dba)_2_/*rac*-BINAP (1/1.5 mol%) (dba = dibenzylidene acetone, *rac*-BINAP = 2,2′-bis(diphenylphosphino)-1,1′-binaphthalene), one of the most efficient catalytic systems for amination, was used for this purpose. Sodium *tert*-butoxide was taken as a base and the reactions were run in boiling dioxane. The arylation of the trioxadiamine **1** with 4- and 3-bromofluorobenzenes proceeded normally to give high yields of the diarylation products **4** and **5** (Table 1, entries 15 and 16), but 1 mol% catalyst was insufficient in the case of 2-bromofluorobenzene. The increase of the catalyst loading to 4 mol% afforded the product **6** in near quantitative yield (entry 17). For dioxadiamine **2** 1 mol% catalyst was also not sufficient even for the reaction with 4-bromofluorobenzene, thus we employed 2 mol% for this process and isolated the target diaryl derivative **7** together with a small amount of the monoaryl substituted diamine **7a** (entry 18). The yield of di (3-fluorophenyl) derivative **8** was as high as 98% under these conditions (entry 19), and surprisingly it was found that the reaction with the *ortho*-isomer afforded 63% of the target compound **9** when applying only 1 mol% catalyst (entry 20). Finally, this amount of the catalyst was found to be sufficient in the diarylation of the dioxadiamine **3** though the yield of di(2-fluorophenyl) derivative **12** was moderate (entries 21–23). Less significant data on the peculiarities of the copper- and palladium-catalyzed arylation of oxadiamines with iodo- and bromofluorobenzenes are to be found in Appendix A.

### 2.2. Catalytic Amination of Halogeno(Trifluoromethyl)Benzenes

At the next step, we studied the Cu(I)-catalyzed arylation of the oxadiamines with iodo(trifluoromethyl)benzenes. For this purpose, isomeric 4- and 3-iodo(trifluoromethyl)benzenes and the compound with additional electron withdrawing group—4-iodo-2-(trifluoromethyl)benzonitrile—were studied (Scheme 2). The results of the reactions are given in Table 2. In all studied processes the application of the L1 ligand was advantageous over the use of the ligand L2 (Table 2, entry 2), it corresponds with the data obtained with isomeric fluoroiodobenzenes. Although the reactions of both iodo(trifluoromethyl)benzenes with the trioxadiamine **1** provided poor yields of the corresponding products **14** and **15** (entries 1 and 3), oxadiamines 2 and 3 gave excellent yields over 90% with the *meta*-isomer (entries 5 and 7), but the *para*-isomer provided insufficient results (entries 4 and 6). Judging from the NMR spectra of the reactions mixtures, it occurred obviously due to concurrent catalytic reactions other than amination which are possible with this reactive aryl iodide. No products of monoarylation could be isolated in individual state in these reactions. The application of 4-iodo-2-(trifluoromethyl)benzonitrile either did not lead to a high yield of the target diarylated product **20** (17%). This was due to the *N*,*N*-diarylation process, which is enough rare for Cu(I)-catalyzed amination and is much more common for Pd(0)-mediated amination [30]. The isolated by-products and their yields under various conditions are to be found in SI. Note that the change of Cu(I) catalyst for Pd(0) catalyst in the above-mentioned reactions cannot be regarded as a solution of the problem. We tried to proceed the arylation of trioxadiamine 1 with the isomeric 4- and 3-trifluoroiodobenzenes in the presence of Pd(dba)_2_/BINAP and, in all cases, the results were even worse.

Having faced problems with the copper- and palladium-mediated amination of iodo(trifluoromethyl)benzenes, we conducted a series of experiments employing bromobenzenes bearing fluorine and trifluoromethyl substituents in Pd(0)-catalyzed amination using trioxadiamine **1** (Scheme 3). The use of 1/1.5 mol% of Pd(dba)_2_/BINAP catalytic system afforded 80% yield of the target product **21** in the reaction with spatially unhindered 1-bromo-3,5-di(trifluoromethyl)benzene, and the triarylated by-product **21a** was also isolated in a small amount (Table 2, entry 8). An increase in the bulkiness of the substituents at the bromine atom in isomeric 1-bromo-2-fluoro-3-(trifluoromethyl)benzene and 2-bromo-1-fluoro-4-(trifluoromethyl)benzene led to insignificant decrease in the yields of corresponding derivatives **22** and **23** (entries 9 and 10). The introduction of more sterically hindered compounds in which the trifluoromethyl group is in the *ortho*-position to bromine demanded an increase of the catalyst to 8 mol% and the products **24**–**26** were obtained in low yields (entries 11–13). The analysis of the reaction mixtures by ^1^H NMR showed that many side processes other than amination took place in these cases (see also Appendix A).

### 2.3. Cu(I)- and Pd(0)-Catalyzed Arylation of Trioxadiamine 1 with Aryl Iodides and Aryl Bromides Bearing Electron Withdrawing and Electron Donor Substituents

The results discussed above demonstrate that in many cases the fluorine-containing substituents did not provide high yields of the *N*,*N′*-dirylated products. To ensure a deeper insight in the scope and limitations of the catalytic diarylation of oxadiamines, we carried out an additional investigation englobing a variety of iodo- and bromobenzenes possessing selected electron-withdrawing (cyano, acetyl, benzoyl, and phenyl) and electron-donating (methoxy) groups. The reactions were run with trioxadiamine **1**, and for copper-catalyzed amination, a more efficient CuI/L1 (20/40 mol%) catalytic system was employed, whereas in the palladium-catalyzed reactions, a Pd(dba)_2_/BINAP (1/1.5 mol%) catalytic system was tested (Scheme 4).

The data collected demonstrate that in the case of electron-withdrawing substituents, better yields of the diarylated products were obtained with *para*-substituted iodobenzenes (Table 3, entries 1, 4, 6, and 7), whereas *meta*-isomers provided somewhat lower yields (entries 2, 5, and 8), which is expected. The best yield (91%) was obtained in the reaction with 4-iodobenzonitrile and with less accepting substituents yields were lower. In the case of a more spatially hindered 2-iodobenzonitrile, the products of di- and monoarylation (**29** and **29a**) were obtained almost in equal amounts (entry 3). On the other hand, in the case of the electron donor methoxy substituent, the reaction with 3-iodoanisole was more efficient affording diarylated product **36** in 64% yield (entry 10), whereas with the *para*-isomer, the target diaryl derivative **35** was formed in only 43% yield (entry 9).

The possibility of obtaining the same products was also studied using Pd(0)-catalyzed reactions. The reaction with 4-bromobenzonitrile was quite successful, providing target compound **27** in 84% yield (Table 3, entry 11). However, 1 mol% catalyst was insufficient for normal diarylation with less reactive 3- and 2-iodobenzonitriles. The application of 4 mol% catalyst solved the problem, and corresponding diaryl derivatives **28** and **29** were obtained in 94 and 40% yields, respectively (entries 12 and 13). Bromoacetophenones were unstable under the reaction conditions, probably due to the action of *t*BuONa, thus the only possibility to introduce these substituents in the diamines is the application of copper-catalyzed reactions. On the other hand, stable bromobenzophenones allowed the synthesis of corresponding products **32** and **37** in 96 and 82% yields, respectively (entries 14 and 19). Note that the Pd(0)-catalyzed reaction of 4-bromobiphenyl provided the same 65% yield of the compound **33** (entry 15) as the above-mentioned Cu(I)-catalyzed amination of 4-iodobiphenyl (entry 7). Diarylation with the isomeric 3-bromobiphenyl was also successful, giving 79% yield of the product **34** (entry 16). The results of the reactions with 4- and 3-bromoanisoles were quite similar to copper-catalyzed ones with 4- and 3-iodoanisoles: in the case of 4-bromoanisole, the yield of **35** was 41% (entry 17), whereas with 3-bromoanisole, in which the electron donor nature of methoxy group does not alter much the reactivity of the bromine atom, the yield of **36** increased to 73% (entry 18).

We have observed an unusual side reaction that diminished the yields of the target compounds—the formation of amino alcohols and diols in Cu(I)-catalyzed amination reactions. Currently, it is too premature to propose a mechanism for the Cu(I)-catalyzed C–O bond cleavage, and we shall not discuss the details; more experimental information is to be found in the SI (Appendix A).

## 3. Materials and Methods

All starting materials purchased from Sigma-Aldrich and Fluka chemical companies were used without further purification. ^1^H, ^13^C, and ^19^F NMR spectra were registered with Bruker Avance 400 and Agilent 400 MR spectrometers (400, 100.6, and 376.4 MHz, respectively) in CDCl_3_ at 298K using residual peaks of the solvent as standards. MALDI-TOF mass spectra were registered with Bruker Autoflex II mass spectrometer in positive mode using dithranol as matrix and poly(ethylene)glycols as internal standards. Preparative column chromatography was performed using silica gel from Merck Co (40/60). The syntheses under the same conditions were conducted using Radleys Carousel 12 Plus reaction station. Pd(dba)_2_ was obtained via a procedure described in [31].

General method (A) for Cu(I)-catalyzed *N*,*N*′-diarylation of oxadiamines. The reaction vessel was flushed with dry argon, equipped with a magnetic stirrer and reflux condenser, and charged with CuI and the ligand (2-isobutyrylcyclohexanone or *rac*-BINOL); 1.25 mmol of the appropriate aryl halide and dry DMF (1 mL) were added followed by the oxadiamine (0.5 mmol). The reaction mixture was stirred for several minutes, then Cs_2_CO_3_ (1.25 mmol) was added and the reaction mixture was stirred at 140 °C for 24 h to ensure the full completion of the process. Next, the reaction mixture was cooled to ambient temperature, a small amount of the solution was taken for ^1^H NMR investigation of its composition, dichloromethane (5–10 mL) was added, the residue filtered off washed with dichloromethane (5–10 mL), and the combined organic fractions were evaporated in vacuo and chromatographed on silica gel using a sequence of eluents: CH_2_Cl_2_, CH_2_Cl_2_-MeOH (200:1, 100:1, 50:1, 20:1, 10:1).

General method (B) for Pd(0)-catalyzed *N,N′*-diarylation of oxadiamines. The reaction vessel was flushed with dry argon, equipped with a magnetic stirrer and reflux condenser, and charged with Pd(dba)_2_ and BINAP ligand, corresponding to aryl halide (1–1.25 mmol) and 5 mL absolute dioxane. After stirring the mixture for several minutes, the appropriate oxadiamine (0.5 mmol) and *t*BuONa (1.5 mmol) were added. The reaction mixture was refluxed for 8 h, and its work up was essentially the same as described for Cu(I)-catalyzed arylation.

*N,N′-(((oxybis(ethane-2,1-diyl))bis(oxy))bis(propane-3,1-diyl))bis(4-fluoroaniline)* (**4**). Obtained according to method B from trioxadiamine **1** (0.5 mmol, 110 mg), 4-bromofluorobenzene (1.25 mmol, 218 mg) in the presence of Pd(dba)_2_ (2.9 mg) and BINAP (4.7 mg). Eluent CH_2_Cl_2_–MeOH 100:1. Yield 147 mg (72%). ^1^H-NMR (400 MHz, CDCl_3_) δ 1.86 (quintet, 4H, *^3^J* = 6.1 Hz, C**H**_2_CH_2_N), 3.17 (t, 4H, *^3^J* = 6.4 Hz, CH_2_N), 3.56–3.61 (m, 8H, OCH_2_), 3.64–3.67 (m, 4H, OCH_2_), 4.34 (br. s, 2H, NH), 6.55 (dd, 4H, ^3^*J_HH_* = 8.8 Hz, *^4^J_HF_*= 4.3 Hz, H2, H2′ (Ph)), 6.84–6.85 (m, 4H, H3, H3′ (Ph)). ^13^C-NMR (100.6 MHz, CDCl_3_) δ 28.6 (2C, **C**H_2_CH_2_N), 42.1 (2C, CH_2_N), 69.4 (2C, OCH_2_), 70.1 (2C, OCH_2_), 70.4 (2C, OCH_2_), 113.6 (d, 4C, ^3^*J_CF_* = 6.6 Hz, C2, C2′(Ph)), 115.1 (d, 4C, ^2^*J_CF_* = 22.1 Hz, C3, C3′(Ph)), 144.5 (2C, C1(Ph)), 155.1 (d, 2C, ^2^*J_CF_* = 235.1 Hz, C4 (Ph)). ^19^F-NMR (376.4 MHz, CDCl_3_) δ –127,30 br. s. MS (MALDI-TOF+): Calculated for C_22_H_31_F_2_N_2_O_3_ [M + H] 409.2303, found 409.2267.

*N,N′-(((oxybis(ethan-2,1-diyl))bis(oxy))bis(propane-3,1-diyl))bis(3-fluoroaniline)* (**5**). Obtained according to method A from trioxadiamine **1** (0.5 mmol, 110 mg), 3-fluoroiodobenzene (1.25 mmol, 278 mg) in the presence of CuI (19 mg) and 2-isobutyrylcyclohexanone (34 mg). Eluent CH_2_Cl_2_–MeOH 100:1. Yield 167 mg (82%). ^1^H-NMR (400 MHz, CDCl_3_) δ 1.88 (quintet, 4H, *^3^J* = 6.1 Hz, C**H**_2_CH_2_N), 3.21 (q, 4H, *^3^J* = 6.4 Hz, CH_2_N), 3.59–3.64 (m, 8H, OCH_2_), 3.68–3.70 (m, 4H, OCH_2_), 4.25 (br. s, 2H, NH), 6.32–6.37 (m, 4H, H4, H6 (Ph)), 6.28 (dt, 2H, *^3^J_HF_* = 11.8 Hz, *^4^J_HH_* = 2.3 Hz, H2 (Ph)); 7.06 (td, (2H, *^3^J_HH_* = 8.2 Hz, *^4^J_HF_* = 6.8 Hz, H5 (Ph)). ^13^C-NMR (100.6 MHz, CDCl_3_) δ 28.4 (2 C, **C**H_2_CH_2_N), 41.5 (2C, CH_2_N), 69.4 (2C, OCH_2_), 69.8 (2C, OCH_2_), 70.2 (2 C, OCH_2_), 98.8 (d, 2C, ^2^*J_CF_* = 25.2 Hz, C2(Ph)), 102.8 (d, 2C, ^2^*J_CF_* = 21.6 Hz, C4(Ph)), 108.8 (2C, C6(Ph)), 129.7 (d, 2C, ^3^*J_CF_* = 10.3 Hz, C5(Ph)), 150.0 (d, 2C, ^3^*J_CF_* = 11.1 Hz, C1(Ph)), 163.7 (d, 2C, ^1^*J_CF_* = 241.6 Hz, C3(Ph)). ^19^F-NMR (376.4 MHz, CDCl_3_) δ –113.05 (ddd, *^3^J_HF_ =* 11.8 Hz, *^3^J_HF_ =* 8.9 Hz, *^4^J_HF_ =* 6.8 Hz). MS (MALDI-TOF+): Calculated for C_22_H_31_F_2_N_2_O_3_ [M + H] 409.230, found 409.244.

*N,N′-(((oxybis(ethan-2,1-diyl))bis(oxy))bis(propane-3,1-diyl))bis(2-fluoroaniline)* (**6**). Obtained according to method B from trioxadiamine **1** (0.5 mmol, 110 mg), 2-bromofluorobenzene (1.25 mmol, 218 mg) in the presence of Pd(dba)_2_ (11 mg) and BINAP (14 mg). Eluent CH_2_Cl_2_–MeOH 200:1. Yield 200 mg (98%). ^1^H-NMR (400 MHz, CDCl_3_) δ 1.91 (quintet, 4H, ^3^*J* = 6.1 Hz, CH_2_CH_2_N), 3.25 (t, 4H, ^3^*J* = 6.5 Hz, CH_2_N), 3.59–3.62 (m, 8H, OCH_2_), 3.67–3.69 (m, 4H, OCH_2_), 4.21 (br. s, 2H, NH), 6.58 (ddd, 2H, ^3^*J*_HH_ = 7.8 Hz, ^4^*J*_HF_ = 5.3 Hz, ^4^*J*_HH_ = 1.1 Hz, H4 (Ph)), 6.67–6.71 (m, 2H, H6(Ph)), 6.95 (ddd, 2H, ^3^*J*_HH_ = 8.0 Hz, ^3^*J*_HF_ = 11.9 Hz, ^4^*J*_HH_ = 1.1 Hz, H3(Ph)), 6.98 (t, 2H, ^3^*J* = 8.0 Hz, H5(Ph)). ^13^C-NMR (100.6 MHz, CDCl_3_) δ 28.7 (2C, CH_2_CH_2_N), 41.0 (2C, CH_2_N), 69.3 (2C, OCH_2_), 70.0 (2C, OCH_2_), 70.3 (2C, OCH_2_), 111.5 (2C, C6 (Ph)), 113.8 (d, 2C, ^2^*J_CF_* = 18.6 Hz, C3 (Ph)), 115.8 (d, 2C, ^3^*J_CF_* = 6.8 Hz, C4 (Ph)), 124.2 (2C, C5 (Ph)), 136.5 (d, 2C, ^3^*J_CF_* = 11.6 Hz, C1 (Ph)), 151.2 (d, 2C, ^2^*J_CF_* = 237.9 Hz, C2 (Ph)). ^19^F-NMR (376.4 MHz, CDCl_3_) δ –136.80 (ddd, *^3^J_HF_*= 11.9 Hz, *^4^J_HF_*= 6.7 Hz, *^4^J_HF_*= 5.3 Hz). MS (MALDI-TOF+): Calculated for C_22_H_31_F_2_N_2_O_3_ [M + H] 409.2303, found 409.2345.

*N,N′-((ethane-1,2-diyl(oxy))bis(ethane-2,1-diyl))bis(4-fluoroaniline)* (**7**). Obtained according to method B from dioxadiamine **2** (0.5 mmol, 74 mg), 4-bromofluorobenzene (1.25 mmol, 218 mg) in the presence of Pd(dba)_2_ (5.7 mg) and BINAP (7.8 mg). Eluent CH_2_Cl_2_–MeOH 200:1. Yield 123 mg (73%). ^1^H-NMR (400 MHz, CDCl_3_) δ 3.26 (t, 4H, ^3^*J* = 5.2 Hz, CH_2_NH), 3.66 (s, 4H, OCH_2_CH_2_O), 3.72 (t, 4H, ^3^*J* = 5.1 Hz, OCH_2_CH_2_N), 3,98 (br.s, 2H, NH), 6.63 (dd, 4H, ^3^*J*_HH_ = 8.8 Hz, ^4^*J*_HF_ = 4.4 Hz, H2, H2′ (Ph)), 6.85–6.90 (m, 4H, H3, H3′ (Ph)). ^13^C-NMR (100.6 MHz, CDCl_3_) δ 43.8 (2C, CH_2_N), 69.1 (2C, OCH_2_CH_2_N), 69.8 (2C, OCH_2_CH_2_O), 113.6 *(*d, 4C, ^3^*J_CF_*= 7.0 Hz, C2, C2′ (Ph)), 115.2 (d, 4C, *^2^J_CF_* = 22.3 Hz, C3, C3′ (Ph)), 144.1 (2 C, C1 (Ph)), 155.5 (d, 2 C, *^1^J_CF_* = 235 Hz, C4 (Ph)). ^19^F-NMR (376.4 MHz, CDCl_3_) δ –136.80 (tt, *^3^J_HF_*= 8.7 Hz, *^4^J_HF_* = 4.4 Hz). MS (MALDI-TOF+): Calculated for C_18_H_23_F_2_N_2_O_2_ [M + H] 337.1728, found 337.1702.

*4-Fluoro-N-(4-fluorophenyl)-N-(2-(2-(2-((4-fluorophenyl)amino)ethoxy)ethoxy)ethyl)aniline* (**7a**). Obtained as the second product in the synthesis of compound **7** using method B. Eluent CH_2_Cl_2_. Yield 24 mg (11%). ^1^H-NMR (400 MHz, CDCl_3_) δ 3.25 (t, 2H, ^3^*J* = 5.2 Hz, CH_2_NHAr), 3.59 (s, 4H, OCH_2_CH_2_O), 3.64 (t, 2H, ^3^*J* = 5.2 Hz, OCH_2_CH_2_NHAr), 3.67 (t, 2H, ^3^*J* = 6.1 Hz, OCH_2_CH_2_NAr_2_), 3.83 (t, 2H, ^3^*J* = 6.1 Hz, CH_2_NAr_2_), 6.62 (dd, 2H, ^3^*J*_HH_ = 8.8 Hz, ^4^*J*_HF_ = 4.3 Hz, H (Ph)), 6.89 (t, 2H, ^3^*J*_HH_ = 8.7 Hz 2H, H (Ph)), 6.92-6.95 (m, 8H, H (Ph)), NH proton was not unambiguously assigned. ^13^C-NMR (100.6 MHz, CDCl_3_) δ 45.8 (1C, CH_2_NHAr), 52.1 (1C, CH_2_NAr_2_), 68.2 (1C, OCH_2_), 68.5 (1C, OCH_2_), 68.5 (1C, OCH_2_), 70.3 (1C, OCH_2_), 70.6 (1C, OCH_2_), 115.8 (d, 6C, ^2^*J*_CF_ = 22.1 Hz, CH (Ph)), 116.4 (br. s, 2C, CH (Ph)), 122.3 (d, 4C, ^3^*J*_CF_ = 7.4 Hz, CH (Ph)), 144.3 (2C, CN (Ph)), 157.9 (d, 2C, ^1^*J*_CF_ = 240.9 Hz, CF (Ph)), two quaternary carbon atoms were not unambiguously assigned. MS (MALDI-TOF+): Calculated for C_24_H_26_F_3_N_2_O_2_ [M + H] 431.195, found 431.184.

*N,N′-((ethane-1,2-diyl(oxy))bis(ethane-2,1-diyl))bis(3-fluoroaniline)* (**8**). Obtained according to method A from dioxadiamine **2** (0.5 mmol, 74 mg), 3-fluoroiodobenzene (1.25 mmol, 278 mg) in the presence of CuI (19 mg) and 2-isobutyrylcyclohexanone (34 mg). Eluent CH_2_Cl_2_–MeOH 200:1. Yield 118 mg (70%). ^1^H-NMR (400 MHz, CDCl_3_) δ 3.26 (t, 4H, ^3^*J* = 5.2 Hz, CH_2_NH), 3.66 (s, 4H, OCH_2_CH_2_O), 3.72 (t, 4H, ^3^*J* = 5.1 Hz, OCH_2_CH_2_N), 4.01 (br. s, 2H, NH), 6.30 (dt, 2H, ^3^*J*_HF_ = 11.5 Hz, ^4^*J*_HH_ = 2.1 Hz, H2 (Ph)), 6.36–6.41 (m, 4H, H4, H6 (Ph)), 7.08 (td, 2H, ^3^*J*_HH_ = 8.0 Hz, ^4^*J*_HF =_ 6.9 Hz, H5 (Ph)). ^13^C-NMR (100.6 MHz, CDCl_3_) δ 43.1 (2C, CH_2_N), 69.0 (2C, OCH_2_CH_2_N), 69.8 (2C, OCH_2_CH_2_O), 99.3 (d, 2C, ^2^*J_CF_* = 25.4 Hz, C2 (Ph)), 103.6 (d, 2C, ^2^*J_CF_* = 21.4 Hz, C4 (Ph)), 108.6 (2C, C6 (Ph)), 129.8 (d, 2C, ^3^*J_CF_* = 10.1 Hz, C5 (Ph)), 149.4 (d, 2C, ^3^*J_CF_* = 10.9 Hz, C1 (Ph)), 163.7 (d, 2C, ^1^*J_CF_* = 242.7 Hz, C3 (Ph)). ^19^F-NMR (376.4 MHz, CDCl_3_) δ –136.80 (ddd, *^3^J_HF_ =* 11.5 Hz, *^3^J_HF_* = 6.9 Hz, *^4^J_HF_* = 2.1 Hz). MS (MALDI-TOF+): Calculated for C_18_H_21_F_2_N_2_O_2_ [*M* – H_2_ + H] 335.1571, found 335.1546. 

*N,N′-((ethane-1,2-diyl(oxy))bis(ethane-2,1-diyl))bis(2-fluoroaniline)* (**9**). Obtained according to method B from dioxadiamine **2** (0.5 mmol, 74 mg), 2-bromofluorobenzene (1.25 mmol, 218 mg) in the presence of Pd(dba)_2_ (2.9 mg) and BINAP (4.7 mg). Eluent CH_2_Cl_2_–MeOH 200:1. Yield 105 mg (63%). ^1^H-NMR (400 MHz, CDCl_3_) δ 3.34 (t, H, ^3^*J* = 5.2 Hz, CH_2_NH,), 3.67(s, 4H, OCH_2_CH_2_O), 3.73 (t, 4H, ^3^*J* = 5.2 Hz, OCH_2_CH_2_N), 4.30 (br. s, 2H, NH), 6.63 (td, 2H, ^3^*J*_HH_ = 7.2 Hz, ^4^*J*_HF_ = 5.7 Hz, H4 (Ph)), 6.70–6.74 (m, 2H, H6 (Ph)), 6.93–7.00 (m, 4H, H3, H5 (Ph)). ^13^C-NMR (100.6 MHz, CDCl_3_) δ 42.9 (2C, CH_2_N), 69.1 (2C, O**C**H_2_CH_2_N), 70.0 (2C, OCH_2_CH_2_O), 112.1 (2C, C6 (Ph)), 114.1 д (2C, ^2^*J*_CF_ = 18.6 Hz, C3 (Ph)), 116.6 d (2C, ^3^*J*_CF_ = 6.4 Hz, C4 (Ph)), 124.1 (2C, C5 (Ph)), 136.1 д (2C, ^2^*J*_CF_ = 11.2 Hz, C1 (Ph)), 151.4 д (2 C, ^2^*J*_CF_ = 239.2 Hz, C2 (Ph)). ^19^F-NMR (376.4 MHz, CDCl_3_) δ –136.10 br. s. MS (MALDI-TOF+): Calculated for C_18_H_21_F_2_N_2_O_2_ [*M* – H_2_+ H] 335.1571, found 335.1552.

*N-(2-(2-(2-aminoethoxy)ethoxy)ethyl)-2-fluoroaniline* (**9a**). Obtained as the second product in the synthesis of compound **9** using method A. Eluent CH_2_Cl_2_–MeOH 50:1. Yield 19 mg (16%). ^1^H-NMR (400 MHz, CDCl_3_) δ 3.32–3.35 (m, 2H, CH_2_N), 3.50 (q, 2H, ^3^*J* = 5.0 Hz, CH_2_N), 3.58 (t, 4H, ^3^*J* = 4.8 Hz, OCH_2_), 3.64–3.66 (m, 2H, OCH_2_), 3.73 (t, 2H, ^3^*J* = 5.2 Hz, OCH_2_), 6.14 (br. s, 1H, NH), 6.62–6.69 (m, 1H, H4(Ph)), 6.73 (dd, 1H, ^3^*J*_HF_ = 8.2 Hz, ^3^*J*_HH_ = 8.2 Hz, H6(Ph)), 6.94–7.00 (m, 1H, H3(Ph)), 7.00 (dd, 1H, ^3^*J*_HFobs_ = 8.0 Hz, H5(Ph)), NH_2_ protons were not unambiguously assigned. MS (MALDI-TOF+): Calculated for C_12_H_20_FN_2_O_2_ [*M* + H] 243.1508, found 243.1466.

*N,N′-((butane-1,4-diylbis(oxy))bis(propane-3,1-diyl))bis(4-fluoroaniline)* (**10**). Obtained according to method B from dioxadiamine **3** (0.5 mmol, 102 mg), 4-bromofluorobenzene (1.25 mmol, 218 mg) in the presence of Pd(dba)_2_ (2.9 mg) and BINAP (4.7 mg). Eluent CH_2_Cl_2_–MeOH 200:1. Yield 118 mg (60%). ^1^H-NMR (400 MHz, CDCl_3_) δ 1.65–1.67 (m, 4H, OCH_2_**CH_2_CH_2_**CH_2_O), 1.87 (quintet, 4H, *^3^J* = 6.0 Hz, OCH_2_**CH_2_**CH_2_N), 3.17 (t, 4H, ^3^*J* = 6.5 Hz, CH_2_N), 3.43–3.46 (m, 4H, OC**H**_2_CH_2_CH_2_C**H**_2_O), 3.53 (t, 4H, ^3^*J* = 5.8 Hz, OC**H**_2_CH_2_CH_2_N), 4.09 (br. s, 2H, NH), 6.54 (dd, 4H, ^3^*J*_HH_ = 8.8 Hz, ^4^*J*_HF_ = 4.4 Hz, H2, H2′ (Ph)), 7.85–7.89 (m, 4H, H3, H3′ (Ph)). ^13^C-NMR (100.6 MHz, CDCl_3_) δ 26.2 (2C, CH_2_**C**H_2_**C**H_2_CH_2_), 28.8 (2C, OCH_2_**C**H_2_CH_2_NH), 42.6 (2C, CH_2_N), 69.2 (2C, OCH_2_), 70.4 (2C, OCH_2_), 113.3 (d, 4C, ^3^*J*_CF_ = 7.2 Hz, C2, C2′ (Ph)), 115.2 (d, 4C, ^2^*J*_CF_ = 22.3 Hz, C3, C3′ (Ph)), 144.2 (2C, C1 (Ph)), 155.4 (d, 2C, ^1^*J*_CF_ = 234.4 Hz, C4 (Ph)). ^19^F-NMR (376.4 MHz, CDCl_3_) δ –126.97 br. s. MS (MALDI-TOF+): Calculated for C_22_H_31_F_2_N_2_O_2_ [*M* + H] 393.2354, found 393.2386.

*N,N′-((butane-1,4-diylbis(oxy))bis(propane-3,1-diyl))bis(3-fluoroaniline)* (**11**). Obtained according to method A from dioxadiamine **3** (0.5 mmol, 102 mg), 3-fluoroiodobenzene (1.25 mmol, 278 mg) in the presence of CuI (19 mg) and 2-isobutyrylcyclohexanone (34 mg). Eluent CH_2_Cl_2_–MeOH 200:1. Yield 151 mg (77%). ^1^H-NMR (400 MHz, CDCl_3_) δ 1.69–1.71 (m, 4H, OCH_2_C**H**_2_C**H**_2_CH_2_O), 1.88 (quintet, 4H, ^3^*J* = 6.1 Hz, OCH_2_C**H**_2_CH_2_N), 3.25 (t, 4H, ^3^*J* = 6.3 Hz, CH_2_N), 3.46–3.48 (m, 4H, OCH_2_), 3.55 (t, 4H, ^3^*J* = 5.7 Hz, OCH_2_), 4.20 (br. s, 2H, NH), 6.32 (dt, 2H, ^3^*J*_HF_ = 11.8 Hz, ^4^*J*_HH_ = 2.2 Hz, H2 (Ph)), 6.36–6.40 (m, 4 H, H4, H6 (Ph)), 7.10 (td, 2H, ^3^*J*_HH_ = 7.6 Hz, ^4^*J*_HF_ = 7.6 Hz, H5 (Ph)). ^13^C-NMR (100.6 MHz, CDCl_3_) δ 26.2 (2C, CH_2_**C**H_2_**C**H_2_CH_2_), 28.8 (2C, OCH_2_**C**H_2_CH_2_NH), 41.6 (2C, CH_2_N), 69.0 (2C, OCH_2_), 70.5 (2C, OCH_2_), 98.7 (d, 2C, ^2^*J*_CF_ = 25.3 Hz, C2 (Ph)), 102.8 (d, 2C, ^2^*J*_CF_ = 21.6 Hz, C4 (Ph)), 108.2 (2C, C6 (Ph)), 129.8 (d, 2C, ^3^*J*_CF_ = 10.1 Hz, C5 (Ph)), 150.0 (d, 2C, ^3^*J*_CF_ = 7.7 Hz, C1 (Ph)), 163.8 (d, 2C, ^1^*J*_CF_ = 241.8 Hz, C3 (Ph)). ^19^F-NMR (376.4 MHz, CDCl_3_) δ –113.10 (ddd, *^3^J_HF_*= 11.8 Hz, *^3^J_HF_* = 8.7 Hz, *^4^J_HF_* = 7.6 Hz). MS (MALDI-TOF+): Calculated for C_22_H_31_F_2_N_2_O_2_ [*M* + H] 393.2354, found 393.2329.

*N,N′-((butane-1,4-diylbis(oxy))bis(propane-3,1-diyl))bis(2-fluoroaniline)* (**12**). Obtained according to method B from dioxadiamine **3** (0.5 mmol, 102 mg), 2-bromofluorobenzene (1.25 mmol, 218 mg) in the presence of Pd(dba)_2_ (2.9 mg) and BINAP (4.7 mg). Eluent CH_2_Cl_2_–MeOH 200:1. Yield 96 mg (49%). ^1^H-NMR (400 MHz, CDCl_3_) δ 1.67–1.70 (m, 4H, OCH_2_C**H**_2_C**H**_2_CH_2_O), 1.91 (quintet, 4H, ^3^*J* = 6.1 Hz, OCH_2_C**H**_2_CH_2_N), 3.26 (t, 4H, ^3^*J* = 6.2 Hz, CH_2_N), 3.44–3.47 (m, 4H, OCH_2_), 3.55 (t, 4H, ^3^*J* = 5.8 Hz, OCH_2_), 4.32 (br. s, 2H, NH), 6.58 (tdd, 2H, ^3^*J*_HH_ = 7.6 Hz, ^4^*J*_HF_ = 4.9 Hz, ^4^*J*_HH_ = 1.5 Hz, H4 (Ph)), 6.67–6.72 (m, 2H, H6 (Ph)), 6.95 (ddd, 2H, ^3^*J*_HF_ = 11.9 Hz, ^3^*J*_HH_ = 8.2 Hz, ^4^*J*_HH_ = 1.4 Hz, H3 (Ph)), 6.98 (t, 2H, ^3^*J* = 8.0 Hz, H5 (Ph)). ^13^C-NMR (100.6 MHz, CDCl_3_) δ 26.1 (2C, CH_2_**C**H_2_**C**H_2_CH_2_), 28.9 (2C, OCH_2_**C**H_2_CH_2_NH), 41.3 (2C, CH_2_N), 68.9 (2C, OCH_2_), 70.5 (2C, OCH_2_), 111.4 (2C, C6 (Ph)), 113.8 (d, 2C, ^2^*J*_CF_ = 18.1 Hz, C3 (Ph)), 115.7 (d, 2C, ^3^*J*_CF_ = 6.6 Hz, C4 (Ph)), 124.1 (2C, C5 (Ph)), 136.7 (d, 2C, ^3^*J*_CF_ = 11.6 Hz, C1 (Ph)), 151.1 (d, 2C, ^2^*J*_CF_ = 238.5 Hz, C2 (Ph)). ^19^F-NMR (376.4 MHz, CDCl_3_) δ –136.88 (ddd, *^3^J_HF_* = 11.9 Hz, *^3^J_HF_* = 6.5 Hz, *^4^J_HF_* = 4.9 Hz). MS (MALDI-TOF+): Calculated for C_22_H_31_F_2_N_2_O_2_ [*M* + H] 393.2354, found 393.2332.

*N-(3-(4-(3-aminopropoxy)butoxy)propyl)-2-fluoroaniline* (**12a**). Obtained as the second product in the synthesis of compound **12** using method A. Eluent CH_2_Cl_2_–MeOH 50:1. Yield 9126 mg (8%). ^1^H-NMR (400 MHz, CDCl_3_) δ 1.63–1.68 (m, 4H, OCH_2_C**H**_2_C**H**_2_CH_2_O), 1.78 (quintet, 2H, ^3^*J* = 6.0 Hz, OCH_2_C**H**_2_CH_2_NH_2_), 1.91 (quintet, 2H, ^3^*J* = 6.0 Hz, OCH_2_C**H**_2_CH_2_NH), 3.25 (t, 2H, ^3^*J* = 5.9 Hz, CH_2_N), 3.38–3.46 (m, 6H, CH_2_N, OCH_2_), 3.51 (t, 2H, ^3^*J* = 5.7 Hz, OCH_2_), 3.56 (t, 2H, ^3^*J* = 5.7 Hz, OCH_2_), 6.22 (br. s, 1H, NH), 6.56–6.62 (m, 1H, H4 (Ph)), 6.69 (dd, 1H, ^3^*J*_HH_ = 8.3 Hz, ^4^*J*_HF_ = 8.3 Hz, H6 (Ph)), 6.94 (dd, 1H, ^3^*J*_HF_ = 12.0 Hz, ^3^*J*_HH_ = 8.1 Hz, H3 (Ph)), 6.98 (t, 1H, ^3^*J*_obs_ = 7.7 Hz, H5 (Ph)). NH_2_ protons were not unambiguously assigned. MS (MALDI-TOF+): Calculated for C_16_H_28_FN_2_O_2_ [*M* + H] 299.2135, found 299.2114.

*N,N′-(((oxybis(ethane-2,1-diyl))bis(oxy))bis(propane-3,1-diyl))bis(2,4-difluoroaniline)* (**13**). Obtained according to method A from trioxadiamine **1** (0.5 mmol, 110 mg), 2,4-difluoro-1-iodobenzene (1.25 mmol, 300 mg) in the presence of CuI (19 mg) and 2-isobutyrylcyclohexanone (34 mg). Eluent CH_2_Cl_2_–MeOH 200:1. Yield 45 mg (20%). ^1^H-NMR (400 MHz, CDCl_3_) δ 1.89 (quintet, 4H, ^3^*J* = 6.0 Hz, OCH_2_C**H**_2_CH_2_NH), 3.21 (t, 4H, ^3^*J* = 6.3 Hz, CH_2_N), 3.58–3.60 (m, 8H, OCH_2_), 3.65–3.67 (m, 4H, OCH_2_), 6.63 (td, 2H, ^3^*J*_HF_ = 9.0 Hz, ^3^*J*_HF_ = 5.8 Hz, H3 (Ph)), 6.70–6.77 (m, 4H, H5, H6 (Ph)), NH were not unambiguously assigned. ^13^C-NMR (100.6 MHz, CDCl_3_) δ 28.5 (2C, OCH_2_**C**H_2_CH_2_NH), 42.2 (2C, CH_2_N), 69.3 (2C, OCH_2_), 70.0 (2C, OCH_2_), 70.2 (2C, OCH_2_), 103.0 (t, 2C, ^2^*J*_CFobs_ = 24.0 Hz, C3 (Ph)), 110.1 (d, 2C, ^2^*J*_CF_ = 22.7 Hz, C5 (Ph)), 112.3 (br. s, 2C, C6 (Ph)), quaternary carbon atoms were not unambiguously assigned due to low intensity of their multiplets. ^19^F-NMR (376.4 MHz, CDCl_3_) δ –107.02 (qd, 2F, *^3^J_HF_* = *^4^J_FF =_* 9.0 Hz, *^4^J_HF_* = 6.3 Hz 4-F), (−115.35)–(−115.43) (m, 2F, 2-F). MS (MALDI-TOF+): Calculated for C_22_H_29_F_4_N_2_O_3_ [*M* + H] 445.2114, found 445.2139.

*N-(3-(2-(2-(3-aminopropoxy)ethoxy)ethoxy)propyl)-2,4-difluoroaniline* (**13a**). Obtained as the second product in the synthesis of compound **13** using method A. Eluent CH_2_Cl_2_–MeOH 50:1. Yield 32 mg (19%). ^1^H-NMR (400 MHz, CDCl_3_) δ 1.76 (quintet, 2H, ^3^*J* = 5.7 Hz, OCH_2_C**H**_2_CH_2_NH_2_), 1.89 (quintet, 2H, ^3^*J* = 5.7 Hz, OCH_2_C**H**_2_CH_2_NHAr), 3.21 (t, 2H, ^3^*J* = 5.1 Hz, C**H**_2_NH), 3.39 (q, 2H, ^3^*J* = 5.4 Hz, CH_2_N), 3.50–3.67 (m, 12 H, OCH_2_), 6.55–6.63 (m, 1H, H3(Ph)), 6.69–6.76 (m, 2H, H5, H6 (Ph)), NH and NH_2_ protons were not unambiguously assigned. MS (MALDI-TOF+): Calculated for C_16_H_27_F_2_N_2_O_3_ [*M* + H] 333.1990, found 333.1964.

*N,N′-(((oxybis(ethane-2,1-diyl))bis(oxy))bis(propane-3,1-diyl))bis(4-(trifluoromethyl)aniline)* (**14**). Obtained according to method A from trioxadiamine **1** (0.5 mmol, 110 mg), 1-iodo-4-(trifluoromethyl)benzene (1.25 mmol, 340 mg) in the presence of CuI (19 mg) and 2-isobutyrylcyclohexanone (34 mg). Eluent CH_2_Cl_2_–MeOH 100:1. Yield 64 mg (25%). ^1^H-NMR (400 MHz, CDCl_3_) δ 1.87 (quintet, 4H, ^3^*J* = 6.0 Hz, C**H**_2_CH_2_N), 3.24 (t, 4H, ^3^*J* = 6.4 Hz, CH_2_N), 3.57–3.62 (m, 8H, OCH_2_), 3.66–3.69 (m, 4H, OCH_2_), 4.60 (br. s, 2H, NH), 6.56 (d, 4H, ^3^*J*_obs_ = 8.6 Hz, H2, H2′ (Ph)), 7.36 (d, 4H, ^3^*J*_obs_ = 8.6 Hz, H3, H3′ (Ph)). ^13^C-NMR (100.6 MHz, CDCl_3_) δ 28.7 (2C, **C**H_2_CH_2_N), 41.5 (2C, CH_2_N), 69.7 (2C, OCH_2_), 70.2 (2C, OCH_2_), 70.6 (2C, OCH_2_), 113.6 (4C, C2, C2′ (Ph)), 118.2 (q, 2C, *^2^J_CF_*= 32.0 Hz, C4 (Ph)), 122.4 (q, 2C, *^1^J_CF_* = 270.6 Hz, CF_3_), 126.7 (4C, C3, C3′ (Ph)), 150.9 (2C, C1 (Ph)). ^19^F-NMR (376.4 MHz, CDCl_3_) δ –61.13 (6F, CF_3_). MS (MALDI-TOF+): Calculated for C_24_H_30_F_5_N_2_O_3_ [*M*–F] 489.2177, found 489.2198.

*N,N′-(((oxybis(ethane-2,1-diyl))bis(oxy))bis(propane-3,1-diyl))bis(3-(trifluoromethyl)aniline)* (**15**). Obtained according to method A from trioxadiamine **1** (0.5 mmol, 110 mg), 1-iodo-3-(trifluoromethyl)benzene (1.25 mmol, 340 mg) in the presence of CuI (19 mg) and 2-isobutyrylcyclohexanone (34 mg). Eluent CH_2_Cl_2_–MeOH 200:1. Yield 53 mg (21%). ^1^H-NMR (400 MHz, CDCl_3_) δ 1.87 (quintet, 4H, ^3^*J* = 5.9 Hz, C**H**_2_CH_2_N), 3.23 (t, 4H, ^3^*J* = 6.3 Hz, CH_2_N), 3.58–3.61 (m, 8H, OCH_2_), 3.66–3.69 (m, 4H, OCH_2_), 4.22 (br. s, 2H, NH), 6.71 (d, 2H, ^3^*J* = 6.6 Hz, H6 (Ph)), 6.77 (s, 2H, H2 (Ph)), 6.88 (d, 2H, ^3^*J* = 8.0 Hz, H4 (Ph)), 7.21 (t, 2H, ^3^*J*_obs_ = 8.0 Hz, H5 (Ph)). ^13^C-NMR (100.6 MHz, CDCl_3_) δ 28.7 (2C, **C**H_2_CH_2_NH), 41.8 (2C, CH_2_N), 69.8 (2C, OCH_2_), 70.2 (2C, OCH_2_), 70.5 (2C, OCH_2_), 108.7 (2C, C2 (Ph)), 113.2 (2C, C4 (Ph)), 115.6 (2C, C6 (Ph)), 124.4 (q, 2C, ^1^*J*_CF_ = 272.2 Hz, CF_3_), 129.5 (2C, C5 (Ph)), 131.3 (q, 2C, ^2^*J*_CF_ = 32.4 Hz, C3 (Ph)), 148.6 (2C, C1 (Ph)). ^19^F-NMR (376.4 MHz, CDCl_3_) δ –62.85 (6F, CF_3_). MS (MALDI-TOF+): Calculated for C_24_H_31_F_6_N_2_O_3_ [*M* + H] 509.224, found 509.233.

*N,N′-((ethane-1,2-diyl(oxy))bis(ethane-2,1-diyl))bis(4-(trifluoromethyl)aniline)* (**16**). Obtained according to method A from dioxadiamine **2** (0.5 mmol, 74 mg), 1-iodo-4-(trifluoromethyl)benzene (1.25 mmol, 340 mg) in the presence of CuI (19 mg) and 2-isobutyrylcyclohexanone (34 mg). Eluent CH_2_Cl_2_–MeOH 200:1. Yield 31 mg (14%). ^1^H-NMR (400 MHz, CDCl_3_) δ 3.32 (t, 4H, ^3^*J* = 5.3 Hz, CH_2_NH), 3.66 (s, 4H, OCH_2_CH_2_O), 3.71 (t, 4 H, ^3^*J* = 5.3 Hz, OCH_2_), 3.93 (br. s, 2H, NH), 6.61 (d, 4H, ^3^*J* = 8.5 Hz, H2, H2′ (Ph)), 7.37 (d, 4H, ^3^*J* = 8.5 Hz, H3, H3′ (Ph)). ^13^C-NMR (100.6 MHz, CDCl_3_) δ 43.1 (2C, CH_2_N), 69.2 (2C, OCH_2_), 70.2 (2C, OCH_2_), 112.2 (4C, C2, C2′ (Ph)), 119.2 (q, 2C, ^2^*J*_CF_ = 32.0 Hz, C4 (Ph)), 122.2 (q, 2C, ^1^*J*_CF_ = 279.1 Hz, CF_3_ (Ph)), 126.6 (br. s, 4C, C3, C3′ (Ph)), 150.3 (2C, C1 (Ph)). ^19^F-NMR (376.4 MHz, CDCl_3_) δ –61.19 (6F, CF_3_). MS (MALDI-TOF+): Calculated for C_20_H_22_F_5_N_2_O_2_ [*M* – F] 417.1601, found 417.1581.

*N,N′-((ethane-1,2-diyl(oxy))bis(ethane-2,1-diyl))bis(3-(trifluoromethyl)aniline)* (**17**). Obtained according to method A from dioxadiamine **2** (0.5 mmol, 74 mg), 1-iodo-3-(trifluoromethyl)benzene (1.25 mmol, 340 mg) in the presence of CuI (19 mg) and 2-isobutyrylcyclohexanone (34 mg). Eluent CH_2_Cl_2_–MeOH 200:1. Yield 214 mg (98%). ^1^H-NMR (400 MHz, CDCl_3_) δ 3.30 (t, 4H, ^3^*J* = 5.2 Hz, CH_2_N), 3.66 (s, 4H, OCH_2_CH_2_O), 3.71 (t, 4H, ^3^*J* = 5.2 Hz, CH_2_O), 4.22 (br. s, 2H, NH), 6.73 (dd, 2H, ^3^*J* = 8.0 Hz, ^4^*J* = 2.0 Hz, H6(Ph)), 6.81 (s, 2H, H2(Ph)), 6.93 (d, 2H, ^3^*J* = 7.6 Hz, H4(Ph)), 7.22 (t, 2H, ^3^*J*_obs_ = 7.9 Hz, H5 (Ph)). ^13^C-NMR (100.6 MHz, CDCl_3_) δ 43.2 (2C, CH_2_N), 69.3 (2C, OCH_2_), 70.2 (2C, OCH_2_), 108.9 (2C, C2(Ph)), 113.6 (2C, C4(Ph)), 116.0 (2C, C6(Ph)), 124.3 (q, 2C, ^1^*J*_CF_ = 272.0 Hz, CF_3_), 129.5 (2C, C5(Ph)), 131.4 (q, 2C, ^2^*J*_CF_ = 31.9 Hz, C3(Ph)), 148.3 (2C, C1 (Ph)). ^19^F-NMR (376.4 MHz, CDCl_3_) δ –62.87 (6F, CF_3_). MS (MALDI-TOF+): Calculated for C_20_H_22_F_6_N_2_O_2_ [*M* + H] 437.1664, found 437.1687.

*N,N′-((butane-1,4-diylbis(oxy))bis(propane-3,1-diyl))bis(4-(trifluoromethyl)aniline)* (**18**). Obtained according to method A from dioxadiamine **3** (0.5 mmol, 102 mg), 1-iodo-4-(trifluoromethyl)benzene (1.25 mmol, 340 mg) in the presence of CuI (19 mg) and 2-isobutyrylcyclohexanone (34 mg). Eluent CH_2_Cl_2_–MeOH 100:1. Yield 61 mg (28%). ^1^H-NMR (400 MHz, CDCl_3_) δ 1.67–1.70 (m, 4H, CH_2_C**H**_2_C**H**_2_CH_2_), 1.94 (quintet, 4H, ^3^*J* = 5.9 Hz, OCH_2_C**H**_2_CH_2_N), 3.33 (t, 4H, ^3^*J* = 6.7 Hz, CH_2_N), 3.47–3.50 (m, 4H, OCH_2_), 3.57 (t, 4H, ^3^*J* = 5.7 Hz, OC**H**_2_CH_2_CH_2_N), 6.88 (d, 4H, ^3^*J*_obs_ = 8.4 Hz, H2, H2′ (Ph)), 7.46 (d, 4H, ^3^*J*_obs_ = 8.4 Hz, H3, H3′ (Ph)), NH protons were not unambiguously assigned. ^13^C-NMR (100.6 MHz, CDCl_3_) δ 26.6 (2C, CH_2_**C**H_2_**C**H_2_CH_2_), 29.0 (2C, OCH_2_**C**H_2_CH_2_N), 41.8 (2C, CH_2_N), 69.5 (2C, OCH_2_), 70.7 (2C, OCH_2_), 111.6 (4C, C2, C2′ (Ph)), 118.3 (q, 2C, ^2^*J*_CF_ = 32.9 Hz, C4 (Ph)), 122.3 (q, 2C, ^1^*J*_CF_ = 269.8 Hz, CF_3_ (Ph)), 126.5 (br. s, 4C, C3, C3′ (Ph)), 150.9 (2C, C1 (Ph)). ^19^F-NMR (376.4 MHz, CDCl_3_) δ –61.48 (6F, CF_3_). MS (MALDI-TOF+): Calculated for C_24_H_31_F_6_N_2_O_2_ [*M* + H] 493.2290, found 493.2334.

*N,N′-((butane-1,4-diylbis(oxy))bis(propane-3,1-diyl))bis(3-(trifluoromethyl)aniline)* (**19**). Obtained according to method A from dioxadiamine **3** (0.5 mmol, 102 mg), 1-iodo-3-(trifluoromethyl)benzene (1.25 mmol, 340 mg) in the presence of CuI (19 mg) and 2-isobutyrylcyclohexanone (34 mg). Eluent CH_2_Cl_2_. Yield 224 mg (91%). ^1^H-NMR (400 MHz, CDCl_3_) δ 1.67–1.70 (m, 4H, OCH_2_C**H**_2_C**H**_2_CH_2_O), 1.89 (quintet, 4H, ^3^*J* = 5.9 Hz, OCH_2_C**H**_2_CH_2_N), 3.24 (t, 4 H, ^3^*J* = 6.5 Hz, CH_2_NH), 3.45–3.48 (m, 4H, OCH_2_), 3.55 (t, 4H, ^3^*J* = 5.7 Hz, OCH_2_), 4.33 (br. s, 2H, NH), 6.73 (dd, 2H, ^3^*J* = 7.8 Hz, ^4^*J* = 1.9 Hz, H6 (Ph)), 6.77 (s, 2H, H2 (Ph)), 6.89 (d, 2H, ^3^*J* = 7.7 Hz, H4 (Ph)), 7.22 (t, 2H, ^3^*J*_obs_ = 7.8 Hz, H5 (Ph)). ^13^C-NMR (100.6 MHz, CDCl_3_) δ 26.6 (2C, CH_2_**C**H_2_**C**H_2_CH_2_), 29.1 (2C, OCH_2_**C**H_2_CH_2_NH), 42.0 (2C, CH_2_N), 69.5 (2C, OCH_2_), 70.8 (2C, OCH_2_), 108.5 (2C, C2 (Ph)), 113.2 (2C, C4 (Ph)), 115.6 (2 C, C6 (Ph)), 124.4 (q, 2C, ^1^*J*_CF_ = 273.5 Hz, CF_3_), 129.5 (2C, C5 (Ph)), 131.4 (q, 2C, ^2^*J*_CF_ = 31.0 Hz, C3 (Ph)), 148.7 (2 C, C1 (Ph)). ^19^F-NMR (376.4 MHz, CDCl_3_) δ –62.86 (6F, CF_3_). MS (MALDI-TOF+): Calculated for C_24_H_31_F_6_N_2_O_2_ [*M* + H] 493.2290, found 493.2334.

*4,4′-((((Oxybis(ethane-2,1-diyl))bis(oxy))bis(propane-3,1-diyl))bis(azandiyl))bis(2-(trifluoromethyl)-benzonitrile)* (**20**). Obtained as one of several products according to method A from trioxadiamine **1** (0.5 mmol, 110 mg), 4-iodo-2-(trifluoromethyl)benzonitrile (1.25 mmol, 371 mg) in the presence of CuI (19 mg) and 2-isobutyrylcyclohexanone (34 mg). Eluent CH_2_Cl_2_–MeOH 200:1. Yield 47 mg (17%). ^1^H-NMR (400 MHz, CDCl_3_) δ 1.88 (quintet, 4H, ^3^*J* = 6.1 Hz, CH_2_C**H**_2_CH_2_NH), 3.28 (q, 4H, ^3^*J* = 5.2 Hz, CH_2_N), 3.59–3.62 (m, 8H, OCH_2_), 3.66–3.69 (m, 4H, OCH_2_), 6.67 (dd, 1H, ^3^*J* = 8.7 Hz, ^4^*J* = 2.4 Hz, H6 (Ph)), 6.80 (d, 2H, ^4^*J* = 2.4 Hz, H2 (Ph)), 7.48 (d, 2H, ^3^*J* = 8.7 Hz, H5 (Ph)). NH protons were not unambiguously assigned. MS (MALDI-TOF+): Calculated for C_26_H_29_F_6_N_4_O_3_ [*M* + H] 559.214, found 559.227.

*N,N′-(((oxybis(ethane-2,1-diyl))bis(oxy))bis(propane-3,1-diyl))bis(3,5-bis(trifluoromethyl)aniline)* (**21**). Obtained according to method B from trioxadiamine **1** (0.5 mmol, 110 mg), 1-bromo-3,5-di(trifluoromethyl)benzene (1 mmol, 293 mg) in the presence of Pd(dba)_2_ (2.9 mg) and BINAP (4.7 mg). Eluent CH_2_Cl_2_–MeOH 200:1. Yield 257 mg (80%). ^1^H-NMR (400 MHz, CDCl_3_) δ 1.88 (quintet, 4H, ^3^*J* = 6.0 Hz, OCH_2_C**H**_2_CH_2_NH), 3.25 (q, 4H, ^3^*J* = 6.2 Hz, CH_2_N), 3.60–3.63 (m, 8H, OCH_2_), 3.69–3.72 (m, 4H, OCH_2_), 4.92 (br. s, 2H, NH), 6.91 (s, 4H, H2, H6 (Ph)), 7.08 (s, 2H, H4 (Ph)). ^13^C-NMR (100.6 MHz, CDCl_3_) δ 28.3 (2C, OCH_2_**C**H_2_CH_2_NH), 42.2 (2C, CH_2_N), 70.0 (2C, OCH_2_), 70.1 (2C, OCH_2_), 70.4 (2C, OCH_2_), 109.5 (2C, C4 (Ph)), 111.6 (4C, C2, C6 (Ph)), 123.7 (q, 4C, ^1^*J*_CF_ = 272.6 Hz, CF_3_), 132.1 (q, 4C, ^2^*J*_CF_ = 32.4 Hz, C3, C5 (Ph)), 149.0 (2C, C1 (Ph)). ^19^F-NMR (376.4 MHz, CDCl_3_) δ –63.19 (12F, CF_3_). MS (MALDI-TOF+): Calculated for C_26_H_29_F_12_N_2_O_3_ [*M* + H] 645.199, found 645.185.

N-(3,5-bis(trifluoromethyl)phenyl)-N-(3-(2-(2-(3-((3,5-bis(trifluoromethyl)phenyl)amino)- propoxy)ethoxy)ethoxy)propyl)-3,5-bis(trifluoromethyl)aniline (**21a**). Obtained as the second ptoduct in the synthesis of compound **21**. Eluent CH_2_Cl_2_. Yield 30 mg (7%). ^1^H-NMR (400 MHz, CDCl_3_) δ 1.85–1.93 (m, 4H, OCH_2_C**H**_2_CH_2_N), 3.28 (t, 2H, ^3^J = 6.1 Hz, C**H**_2_NH), 3.49 (t, 2H, ^3^J = 6.1 Hz, OCH_2_), 3.60–3.63 (m, 6H, OCH_2_), 3.67–3.70 (m, 4H, OCH_2_), 3.96 (t, 2H, ^3^J = 6.1 Hz, CH_2_NPh_2_), 6.95 (s, 2H, H2, H6 (Ph)), 7.11 (s, 1H, H4 (Ph)), 7.46 (s, 4H, H2, H6 (Ph_2_)), 7.49 (s, 2H, H4 (Ph_2_)), NH proton was not unambiguously assigned. ^13^C-NMR (100.6 MHz, CDCl_3_) δ 27.8 (1C, OCH_2_**C**H_2_CH_2_N), 28.3 (1C, OCH_2_**C**H_2_CH_2_N), 42.5 (1C, CH_2_NHPh), 49.1 (1C, CH_2_NPh_2_), 67.3 (1C, OCH_2_), 70.2 (2C, OCH_2_), 70.3 (1C, OCH_2_), 70.4 (1C, OCH_2_), 70.6 (1C, OCH_2_), 109.8 (1C, C4 (Ph)), 111.8 (2C, C2, C6 (Ph)), 115.8 (2C, C4 (Ph_2_)), 120.6 (4C, C2, C6 (Ph_2_)), 123.0 (q, 6C, ^1^J_CF_ = 273.1 Hz, CF_3_), 132.6 (q, 2C, ^2^J_CF_ = 33.3 Hz, C3, C5 (Ph)), 133. 2 (q, 4C, ^2^J_CF_ = 32.8 Hz, C3, C5 (Ph_2_)), 147.8 (2C, C1 (Ph_2_)), 148.9 (1C, C1 (Ph)). ^19^F-NMR (376.4 MHz, CDCl_3_) δ –63.21 (6F, CF_3_), –63.23 (12F, CF_3_). MS (MALDI-TOF+): Calculated for C_34_H_31_F_18_N_2_O_3_ [M + H] 857.247, found 857.255.

*N,N′-(((oxybis(ethane-2,1-diyl))bis(oxy))bis(propane-3,1-diyl))bis(3-trifluoromethyl-2-fluoroaniline)* (**22**). Obtained according to method B from trioxadiamine **1** (0.5 mmol, 110 mg), 1-bromo-2-fluoro-3-(trifluoromethyl)benzene (1 mmol, 243 mg) in the presence of Pd(dba)_2_ (2.9 mg) and BINAP (4.7 mg). Eluent CH_2_Cl_2_–MeOH 200:1. Yield 177 mg (65%). ^1^H-NMR (400 MHz, CDCl_3_) δ 1.91 (quintet, 4H, ^3^*J* = 6.0 Hz, OCH_2_C**H**_2_CH_2_N), 3.26 (t, 4H, ^3^*J* = 6.3 Hz, CH_2_N), 3.59–3.62 (m, 8H, OCH_2_), 3.65–3.68 (m, 4H, OCH_2_), 4.55 (br. s, 2H, NH), 6.78–6.85 (m, 4H, H4, H6 (Ph)), 7.02 т (2H, ^3^*J* = 8.0 Hz, H5 (Ph)). ^13^C-NMR (100.6 MHz, CDCl_3_) δ 28.8 (2C, OCH_2_**C**H_2_CH_2_NH), 41.7 (2C, CH_2_N), 69.7 (2C, OCH_2_), 70.4 (2C, OCH_2_), 70.6 (2C, OCH_2_), 112.5 (q, 2C, ^3^*J*_CF_ = 4.3 Hz, C4(Ph)), 115.0 (br. s, 2C, C6 (Ph)), 117.4 (qd, 2C, ^2^*J*_CF_ = 33.0 Hz, ^2^*J*_CF_ = 10.7 Hz, C3(Ph)), 120.3 (q, 2C, ^1^*J*_CF_ = 272.0 Hz, CF_3_), 124.2 (d, 2C, ^4^*J*_CF_ = 7.6 Hz C5 (Ph)), 137.7 (d, 2C, ^2^*J*_CF_ = 10.7 Hz, C1 (Ph)), 148.1 (d, 2C, ^1^*J*_CF_ = 248.0 Hz, C2 (Ph)). ^19^F-NMR (376.4 MHz, CDCl_3_) δ –61.13 (d, 6F, *J*_FF_ = 12.3 Hz, CF**_3_**); –138.43 (m, 2F, 2-F). MS (MALDI-TOF+): Calculated for C_24_H_29_F_8_N_2_O_3_ [*M* + H] 545.2050, found 545.2017.

*N,N′-(((oxybis(ethane-2,1-diyl))bis(oxy))bis(propane-3,1-diyl))bis(5-trifluromethyl-2-fluoroaniline)* (**23**). Obtained according to method B from trioxadiamine **1** (0.5 mmol, 110 mg), 2-bromo-1-fluoro-4-(trifluoromethyl)benzene (1 mmol, 243 mg) in the presence of Pd(dba)_2_ (2.9 mg) and BINAP (4.7 mg). Eluent CH_2_Cl_2_–MeOH 200:1. Yield 190 mg (70%). ^1^H-NMR (400 MHz, CDCl_3_) δ 1.92 (quintet, 4H, ^3^*J* = 6.0 Hz, ^3^*J*, OCH_2_C**H**_2_CH_2_NH), 3.27 (t, 4H, ^3^*J* = 6.4 Hz, CH_2_N), 3.59–3.62 (m, 8H, OCH_2_), 3.66–3.69 (m, 4H, OCH_2_), 4.61 (br. s, 2H, NH), 6.82–6.86 (m, 4H, H4, H6 (Ph)), 6.98 (dd, 2H, ^3^*J*_HF_ = 11.0 Hz, ^3^*J*_HH_ = 8.2 Hz, H3 (Ph)). ^13^C-NMR (100.6 MHz, CDCl_3_) δ 28.8 (2C, OCH_2_**C**H_2_CH_2_NH), 41.4 (2C, CH_2_N), 69.7 (2C, OCH_2_), 70.4 (2C, OCH_2_), 70.6 (2C, OCH_2_), 108.1 (2C, C4 (Ph)), 113.0 (2C, C6 (Ph)), 114.2 (d, 2C, ^2^*J*_CF_ = 20.1 Hz, C3 (Ph)), 124.2 (q, 2C, ^1^*J*_CF_ = 271.5 Hz, CF_3_), 127.1 (q, 2C, ^2^*J*_CF_ = 30.8 Hz, C5 (Ph)), 137.4 (2C, C1 (Ph)), 152.8 (d, 2C, ^1^*J*_CF_ = 244.2 Hz, C2 (Ph)). ^19^F-NMR (376.4 MHz, CDCl_3_) δ –62.11 (s, 6F, CF_3_); –131.82 (m, 2F, 2-F). MS (MALDI-TOF+): Calculated for C_24_H_29_F_8_N_2_O_3_ [*M* + H] 545.2050, found 545.2021.

*N,N′-(((oxybis(ethane-2,1-diyl))bis(oxy))bis(propane-3,1-diyl))bis(2-trifluoromethyl-6-fluoroaniline)* (**24**). Obtained according to method B from trioxadiamine **1** (0.5 mmol, 110 mg), 2-bromo-1-fluoro-3-(trifluoromethyl)benzene (1 mmol, 243 mg) in the presence of Pd(dba)_2_ (23.2 mg) and BINAP (28.2 mg). Eluent CH_2_Cl_2_–MeOH 200:1. Yield 49 mg (18%). ^1^H-NMR (400 MHz, CDCl_3_) δ 1.86 (quintet, 4H, ^3^*J* = 6.2 Hz, OCH_2_C**H**_2_CH_2_N), 3.46 (td, 4H, ^3^*J*_HH_ = 6.7 Hz, ^4^*J*_HH_ = 4.5 Hz, CH_2_N), 3.56–3.60 (m, 8H, OCH_2_), 3.63–3.66 (m, 4H, OCH_2_), 4.14 (br. s, 2H, NH), 6.71 (td, 2 H, ^3^*J*_HHobs_ = 8.2 Hz, ^4^*J*_HF_ = 4.5 Hz, H3 (Ph)), 7.11 (dd, 2H, ^3^*J*_HF_ = 12.7 Hz, ^3^*J*_HH_ = 8.1 Hz, H5 (Ph)), 7.22 (t, 2H, ^3^*J*
_HH_ = 7.9 Hz, H4 (Ph)). ^13^C-NMR (100.6 MHz, CDCl_3_) δ 30.4 (2 C, OCH_2_**C**H_2_CH_2_NH), 45.1 (d, 2C, ^4^*J*_CF_ = 10.1 Hz, CH_2_N), 69.5 (2C, OCH_2_), 70.4 (2C, OCH_2_), 70.5 (2C, OCH_2_), 107.4 (q, 2C, ^2^*J*_CF_ = 29.9 Hz, C2 (Ph)), 117.8 (d, 2C, ^3^*J*_CF_ = 7.6 Hz, C4 (Ph)), 120.0 (d, 2C, ^2^*J*_CF_ = 21.2 Hz, C5 (Ph)), 122.2 (br. s, 2C, C3 (Ph)), 124.4 (q, 2C, ^1^*J*_CF_ = 271.8 Hz, CF_3_), 135.6 (d, 2C, ^2^*J*_CF_ = 11.0 Hz, C1 (Ph)), 153.5 (d, 2C, ^1^*J*_CF_ = 242.9 Hz, C6 (Ph)). ^19^F-NMR (376.4 MHz, CDCl_3_) δ –61.08 (d, 6F, *J*_FF_ = 12.3 Hz, CF_3_); –124.47 (m, 2F, 2-F). MS (MALDI-TOF+): Calculated for C_24_H_26_F_7_N_2_O_3_ [*M* – H_2_ – F] 523.1832, found 523.1878.

*N,N′-(((oxybis(ethane-2,1-diyl))bis(oxy))bis(propane-3,1-diyl))bis**(2-trifluoromethyl-6-fluoroaniline)* (**24a**). Obtained as the second product in the synthesis of compound **24**. Eluent CH_2_Cl_2_–MeOH 50:1. Yield 13 mg (7%). ^1^H-NMR (400 MHz, CDCl_3_) δ 1.77 (a, 2H, ^3^*J* = 5.9 Hz, OCH_2_C**H**_2_CH_2_NH_2_), 1.88 (q, 2H, ^3^*J* = 5.7 Hz, OCH_2_C**H**_2_CH_2_NHPh), 3.30 (t, 2H, ^3^*J* = 5.9 Hz, C**H**_2_NH_2_), 3.42–3.64 (m, 14 H, CH_2_NPh_,_ OCH_2_), 4.90 (br. s, 1H, NH), 6.80 (td, 1H, ^3^*J*_HHobs_ = 7.8 Hz, ^4^*J*_HF_ = 4.9 Hz, H4 (Ph)), 7.04 (ddd, 1H, ^3^*J*_HF_ = 11.2 Hz, ^3^*J*_HH_ = 7.8 Hz, ^4^*J*_HH_ = 1.5 Hz, H5 (Ph)), 7.43 (d, 1H, ^3^*J* = 7.8 Hz, H5 (Ph)). NH_2_ were not unambiguously assigned. ^13^C-NMR (100.6 MHz, CDCl_3_) δ 29.3 (1C, OCH_2_**C**H_2_CH_2_NHPh), 31.1 (1C, OCH_2_**C**H_2_CH_2_NH_2_), 38.4 (1C, CH_2_NH_2_), 47.0 (1C, CH_2_NPh), 69.9 (1C, OCH_2_), 70.6 (2C, OCH_2_), 70.8 (2C, OCH_2_), 71.1 (1C, OCH_2_), 107.4 (q, 1C, ^2^*J*_CF_ = 29.9 Hz, C2 (Ph)), 117.7 (d, 1C, ^3^*J*_CF_ = 21.2 Hz, C5 (Ph)), 120.0 (d, 1C, ^2^*J*_CF_ = 7.6 Hz, C4 (Ph)), 125.2 (1 C, C3 (Ph)), 123.7 (q, 1C, ^1^*J*_CF_ = 270.5 Hz, CF_3_), 136.9 (d, 1C, ^2^*J*_CF_ = 12.9 Hz, C1 (Ph)), 155.2 (d, 2C, ^1^*J*_CF_ = 240.0 Hz, C6 (Ph)). MS (MALDI-TOF+): Calculated for C_17_H_27_F_4_N_2_O_3_ [*M* + H] 383.1958, found 383.1940.

*N,N′-(((oxybis(ethane-2,1-diyl))bis(oxy))bis(propane-3,1-diyl))bis(2-trifluoromethyl-3-fluoroaniline)* (**25**). Obtained according to method B from trioxadiamine **1** (0.5 mmol, 110 mg), 1-bromo-2-(trifluoromethyl)-3-fluorobenzene (1 mmol, 243 mg) in the presence of Pd(dba)_2_ (23.2 mg) and BINAP (28.2 mg). Eluent CH_2_Cl_2_. Yield 46 mg (17%). ^1^H-NMR (400 MHz, CDCl_3_) δ 1.92 (quintet, 4H, ^3^*J* = 5.8 Hz, OCH_2_C**H**_2_CH_2_N), 3.24 (q, 4H, ^3^*J* = 5.7 Hz, CH_2_N), 3.59–3.62 (m, 8H, OCH_2_), 3.64–3.67 (m, 4H, OCH_2_), 5.22 (br. s, 2H, NH), 6.37 (dd, 2H, ^3^*J*_HH_ = 8.5 Hz, ^3^*J*_HF_ = 11.3 Hz, H4 (Ph)), 6.37 (d, 2H, ^3^*J* = 8.5 Hz, H6 (Ph)), 7.23 (td, 2H, ^3^*J*_HH_ = 8.2 Hz, ^4^*J*_HF_ = 6.3 Hz, H5 (Ph)). ^13^C-NMR (100.6 MHz, CDCl_3_) δ 28.7 (2C, OCH_2_**C**H_2_CH_2_N), 42.3 (2C, CH_2_N), 68.8 (2C, OCH_2_), 70.4 (2C, OCH_2_), 70.5 (2C, OCH_2_), 101.2 (q, 2C, ^2^*J*_CF_ = 30.0 Hz, C2 (Ph)), 103.5 (d, 2C, ^2^*J*_CF_ = 23.2 Hz, C4 (Ph)), 107.3 (2C, C6 (Ph)), 124.6 (q, 2C, ^1^*J*_CF_ = 273.6 Hz, CF_3_), 133.5 (d, 2C, ^3^*J*_CF_ = 11.8 Hz, C5 (Ph)), 147.5 (2C, C1 (Ph)), 161.5 (d, 2C, ^1^*J*_CF_ = 253.4 Hz, C3 (Ph)). MS (MALDI-TOF+): Calculated for C_24_H_29_F_8_N_2_O_3_ [*M* + H] 545.205, found 545.196.

*N,N′-(((oxybis(ethane-2,1-diyl))bis(oxy))bis(propane-3,1-diyl))bis(2-trifluoromethyl-5-fluoroaniline)* (**26**) Obtained according to method B from trioxadiamine **1** (0.5 mmol, 110 mg), 1-bromo-2-(trifluoromethyl)-5-fluorobenzene (1 mmol, 243 mg) in the presence of Pd(dba)_2_ (23.2 mg) and BINAP (28.2 mg). Eluent CH_2_Cl_2_. Yield 79 mg (29%). ^1^H-NMR (400 MHz, CDCl_3_) δ 1.92 (quintet, 4H, ^3^*J* = 5.9 Hz, OCH_2_C**H**_2_CH_2_N), 3.23 (q, 4H, ^3^*J* = 5.7 Hz, CH_2_N), 3.59–3.62 (m, 8H, OCH_2_), 3.65–3.68 (m, 4H, OCH_2_), 5.02 (br. s, 2H, NH), 6.31–6.38 (m, 4H, H4, H6 (Ph)), 7.36 (dd, 2H, ^3^*J*_HH_ = 8.6 Hz, ^4^*J*_HF_ = 6.3 Hz, H3 (Ph)). ^13^C-NMR (100.6 MHz, CDCl_3_) δ 28.6 (2C, OCH_2_**C**H_2_CH_2_N), 41.8 (2C, CH_2_N), 69.7 (2C, OCH_2_), 70.4 (2C, OCH_2_), 70.5 (2C, OCH_2_), 98.4 (d, 2C, ^2^*J*_CF_ = 26.9 Hz, C6 (Ph)), 102.1 (d, 2C, ^2^*J*_CF_ = 22.8 Hz, C4 (Ph)), 109.3 (qd, 2C, ^2^*J*_CF =_ 30.0 Hz, ^4^*J*_CF_ = 2.2 Hz, C2 (Ph)), 124.9 (q, 2C, ^1^*J*_CF_ = 270.0 Hz, CF_3_), 128.6 (dq, 2C, ^3^*J*_CF_ = 11.4 Hz, ^3^*J*_CF_ = 5.7 Hz, C3(Ph)), 147,9 (dq, 2C, ^3^*J*_CF_ = 12.0 Hz, ^3^*J*_CF_ =1.4 Hz, C1(Ph)), 166.2 (dq, 2C, ^1^*J*_CF_ = 248.1 Hz, ^5^*J*_CF_ = 1.0 Hz, C5(Ph)). ^19^F-NMR (376.4 MHz, CDCl_3_) δ –61.98 (d, 6F, *J*_FF_ = 12.3, CF_3_); –107.62 (br. s, 2F, 2-F). MS (MALDI-TOF+): Calculated for C_24_H_29_F_8_N_2_O_3_ [*M* + H] 545.205, found 545.192.

*N-(3-(2-(2-(3-aminopropoxy)ethoxy)ethoxy)propyl)-2-trifluoromethyl-5-fluoroaniline* (**26a**). Obtained as the second product in the synthesis of compound **26**. Eluent CH_2_Cl_2_–MeOH 50:1. Yield 23 mg (12%). ^1^H-NMR (400 MHz, CDCl_3_) δ 1.81 (quintet, 2H, ^3^*J* = 5.8 Hz, OCH_2_C**H**_2_CH_2_NH_2_), 1.90 (quintet, 2H, ^3^*J* = 5.5 Hz, OCH_2_C**H**_2_CH_2_NHPh), 3.24 (t, 2H, ^3^*J* = 5.5 Hz, C**H**_2_NH_2_), 3.45–3.59 (m, 6H, OCH_2_, CH_2_NPh), 3.60 (s, 4H, OCH_2_), 3.63–3.67 (m, 4H, OCH_2_), 5.02 (br. s, 2H, NH), 6.24 (ddd, 1H, ^3^*J*_HH_ = 8.6 Hz, ^3^*J*_HF_ = 8.6 Hz, ^4^*J*_HH_ = 2.0 Hz, H4 (Ph)), 6.33 (dd, 1H, ^3^*J*_HF_ = 12.2 Hz, ^4^*J*_HH_ = 2.0 Hz, H6 (Ph)), 7.28 (dd, 2H, ^3^*J*_HH_ = 8.2 Hz, ^4^*J*_HF_ = 6.3 Hz, H3 (Ph)). ^13^C-NMR (100.6 MHz, CDCl_3_) δ 28.6 (1C, OCH_2_**C**H_2_CH_2_N), 29.7 (1C, OCH_2_**C**H_2_CH_2_N), 36.8 (1C, CH_2_NH_2_) 41.7 (1C, CH_2_N), 68.5 (1C, OCH_2_), 70.1 (1C, OCH_2_), 70.5 (1C, OCH_2_), 70.6 (1C, OCH_2_), 70.7 (1C, OCH_2_), 71.2 (1C, OCH_2_), 98.2 (d, 1C, ^2^*J*_CF_ = 26.9 Hz, C6 (Ph)), 101.2 (d, 1C, ^2^*J*_CF_ = 22.8 Hz, C4 (Ph)), 129.2 (d, 1C, ^3^*J*_CF_ = 11.6 Hz, C3(Ph)), 150.7 (br.s, 1C, C1(Ph)), 165.6 (d, 2C, ^1^*J*_CF_ = 247.9 Hz, C5(Ph)), quaternary carbon atoms C2 and CF_3_ were not unambiguously assigned. MS (MALDI-TOF+): Calculated for C_17_H_27_F_4_N_2_O_3_ [*M* + H] 383.1958, found 383.1977.

*4,4′-((((Oxybis(ethane-2,1-diyl))bis(oxy))bis(propane-3,1-diyl))bis(azanediyl))dibenzonitrile* (**27**). Obtained according to method A from trioxadiamine **1** (0.5 mmol, 110 mg), 4-iodobenzonitrile (1.25 mmol, 286 mg) in the presence of CuI (19 mg) and 2-isobutyrylcyclohexanone (33 mg). Eluent CH_2_Cl_2_/MeOH 200:1 Yield 192 mg (91%). ^1^H-NMR (400 MHz, CDCl_3_) δ 1.86 (quintet, 4H, ^3^*J* = 6.2 Hz, OCHC**H**_2_CH_2_N), 3.23 (t, 4H, ^3^*J* = 6.3 Hz, CH_2_N), 3.56–3.59 (m, 8H, OCH_2_), 3.64–3.66 (m, 4H, OCH_2_), 5.02 (br. s, 2H, NH), 6.53 (d, 4H, ^3^*J*_obs_ = 8.8 Hz, H2, H2′(Ph)), 7.36 (d, 4H, ^3^*J*_obs_ = 8.8 Hz, H3, H3′(Ph)). ^13^C-NMR (100.6 MHz, CDCl_3_) δ 28.1 (2C, OCH_2_CH_2_CH_2_N), 41.1 (2C, CH_2_N), 69.3 (2C, OCH_2_), 69.7 (2C, OCH_2_), 70.1 (2C, OCH_2_), 97.7 (2C, C1(Ph)), 111.8 (4C, C2, C2′(Ph)), 120.3 (2C, CN (Ph)), 133.2 (4C, C3, C3′ (Ph)), 152.0 (2C, C4 (Ph)). MS (MALDI-TOF+): Calculated for C_24_H_31_N_4_O_3_ [*M* + H] 423.2396, found 423.2425.

*3,3′-((((Oxybis(ethane-2,1-diyl))bis(oxy))bis(propane-3,1-diyl))bis(azanediyl))dibenzonitrile* (**28**). Obtained according to method A from trioxadiamine **1** (0.5 mmol, 110 mg), 3-iodobenzonitrile (1.25 mmol, 286 mg) in the presence of CuI (19 mg) and 2-isobutyrylcyclohexanone (33 mg). Eluent CH_2_Cl_2_/MeOH 50:1 Yield 165 mg (78%). ^1^H-NMR (400 MHz, CDCl_3_) δ 1.83 (quintet, 4H, ^3^*J* = 5.9 Hz, CH_2_C**H**_2_CH_2_N), 3.15 (t, 4H, ^3^*J* = 6.3 Hz, CH_2_N), 3.55–3.58 (m, 8H, OCH_2_), 3.64–3.67 (m, 4H, OCH_2_), 6.71–6.73 (m, 4H, H2, H6(Ph)), 6.84 (d, 2H, ^3^*J* = 7.8 Hz, H4(Ph)), 7.13 (t, 2H, ^3^*J* = 7.8 Hz, H5 (Ph)). NH protons were not unambiguously assigned. ^13^C-NMR (100.6 MHz, CDCl_3_) δ 28.1 (2C, CH_2_**C**H_2_CH_2_N), 41.2 (2C, CH_2_N), 69.4 (2C, OCH_2_), 69.7 (2C, OCH_2_), 70.1 (2C, OCH_2_), 112.4 (2C, C3(Ph)), 114.3 (2C, CH(Ph)), 116.5 (2C, CH(Ph)), 119.3 (2C, CN), 119.5 (2C, CH(Ph)), 129.4 (2C, C5(Ph)), 148.3 (2C, C1(Ph)). MS (MALDI-TOF+): Calculated for C_24_H_31_N_4_O_3_ [*M* + H] 423.2396, found 423.2362.

*2,2′-((((Oxybis(ethane-2,1-diyl))bis(oxy))bis(propane-3,1-diyl))bis(azanediyl))dibenzonitrile* (**29**). Obtained according to method B from trioxadiamine **1** (0.5 mmol, 110 mg), 2-bromobenzonitrile (1 mmol, 182 mg) in the presence of Pd(dba)_2_ (11.6 mg) and BINAP (15.7 mg). Eluent CH_2_Cl_2_/MeOH 100:1 Yield 198 mg (04%). ^1^H-NMR (400 MHz, CDCl_3_) δ 1.89 (quintet, 4H, ^3^*J* = 6.0 Hz, CH_2_C**H**_2_CH_2_N), 3.28 (q, 4H, ^3^*J* = 6.0 Hz, CH_2_N), 3.57–3.61 (m, 8H, OCH_2_), 3.64–3.67 (m, 4H, OCH_2_), 5.03 (br. s, 2H, NH), 6.59 (t, 2H, ^3^*J*_obs_ = 7.2 Hz, H4 (Ph)), 6.63 (d, 2H, ^3^*J* = 8.7 Hz, H6(Ph)), 7.31–7.35 (m, 4H, H3, H5(Ph)). ^13^C-NMR (100.6 MHz, CDCl_3_) δ 27.5 (2C, CH_2_CH_2_CH_2_N), 41.0 (2C, CH_2_N), 69.1 (2C, OCH_2_), 70.0 (4C, OCH_2_), 95.0 (2C, C2(Ph)), 110.0 (2C, CH(Ph)), 115.7 (2C, CH(Ph)), 117.7 (2C, CN), 132.4 (2C, CH(Ph)), 133.8 (2C, CH(Ph)), 150.1 (2C, C1(Ph)). MS (MALDI-TOF+): Calculated for C_24_H_31_N_4_O_3_ [*M* + H] 423.2396, found 423.2375.

*2-((3-(2-(2-(3-Aminorpopxy)ethoxy)ethoxy)propyl)amino)benzonitrile* (**29a**). Obtained as the second product in the synthesis of compound **29** according to method A using trioxadiamine **1** (0.5 mmol, 110 mg), 2-iodobenzonitrile (1.25 mmol, 286 mg) in the presence of CuI (19 mg) and 2-isobutyrylcyclohexanone (33 mg). Eluent CH_2_Cl_2_/MeOH 50:1 Yield 61 mg (38%). ^1^H-NMR (400 MHz, CDCl_3_) δ 1.77 (quintet, 2H, ^3^*J* = 5.9 Hz, OCH_2_CH_2_C**H**_2_NH_2_), 1.89 (quintet, 2H, ^3^*J* = 5.8 Hz, OCH_2_C**H**_2_CH_2_NPh), 3.28 (t, 2H, ^3^*J* = 6.2 Hz, C**H**_2_NH_2_), 3.42 (q, 2H, ^3^*J* = 6.1 Hz CH_2_NPh), 3.58–3.63 (m, 8H, OCH_2_), 3.65–3.68 (m, 2H, OCH_2_), 3.69–3.72 (m, 2H, OCH_2_), 6.61–6.65 (m, 2H, H4, H6 (Ph)), 7.34–7.38 (m, H3, H5 (Ph)). MS (MALDI-TOF+): Calculated for C_17_H_28_N_3_O_3_ [*M* + H] 322.213, found 322.205.

*1,1′-((((Oxybis(ethane-2,1-diyl))bis(oxy))bis(propane-3,1-diyl))bis(azanediyl))bis(4,1-phenylene))bis(ethan-1-one)* (**30**). Obtained according to method A using trioxadiamine **1** (0.5 mmol, 110 mg), 4-iodoacetophenone (1.25 mmol, 308 mg) in the presence of CuI (19 mg) and 2-isobutyrylcyclohexanone (33 mg). Eluent CH_2_Cl_2_/MeOH 100:1 Yield 194 mg (85%). ^1^H-NMR (400 MHz, CDCl_3_) δ 1.86 (quintet, 4H, ^3^*J* = 6.0 Hz, CH_2_C**H**_2_CH_2_N), 2.45 (s, 6H, CH_3_), 3.26 (t, 4H, ^3^*J* = 6.4 Hz, CH_2_N), 3.56–3.59 (m, 8H, OCH_2_), 3.64–3.67 (m, 4H, OCH_2_), 5.07 (br. s, 2H, NH), 6.52 (d, 4H, ^3^*J*_obs_ = 8.8 Hz, H2, H2′ (Ph)), 7.76 (d, 4H, ^3^*J*_obs_ = 8.8 Hz, H3, H3′ (Ph)). ^13^C-NMR (100.6 MHz, CDCl_3_) δ 25.9 (2C, CH_3_), 28.5 (2C, CH_2_**C**H_2_CH_2_NH), 41.4 (2C, CH_2_N), 69.6 (2C, OCH_2_), 70.1 (2C, OCH_2_), 70.2 (2C, OCH_2_), 111.3 (4C, C2, C2′ (Ph)), 126.2 (2C, C4 (Ph)), 130.7 (4C, C3, C3′ (Ph)), 152.3 (2C, C1, (Ph)), 196.1 (2C, CO). MS (MALDI-TOF+): Calculated for C_26_H_37_N_2_O_5_ [*M* + H] 457.2702, found 457.2680.

*1,1′-((((Oxybis(ethane-2,1-diyl))bis(oxy))bis(propane-3,1-diyl))bis(azanediyl))bis(3,1-phenylene))bis(ethan-1-one)* (**31**). Obtained according to method A using trioxadiamine **1** (0.5 mmol, 110 mg), 3-iodoacetophenone (1.25 mmol, 308 mg) in the presence of CuI (19 mg) and 2-isobutyrylcyclohexanone (33 mg). Eluent CH_2_Cl_2_/MeOH 100:1 Yield 103 mg (45%). ^1^H-NMR (400 MHz, CDCl_3_) δ 1.89 (quintet, 4H, ^3^*J* = 6.0 Hz, CH_2_C**H**_2_CH_2_NH), 2.55 (s, 6H, CH_3_) 3.28 (t, 4H, ^3^*J* = 6.4 Hz, CH_2_N), 3.59–3.62 (m, 8H, OCH_2_), 3.67–3.71 (m, 4H, OCH_2_), 6.92 (d, 2H, ^3^*J* = 8.8 Hz, H6(Ph)), 7.21–7.32 (m, 6H, H2, H4, H5 (Ph)), NH were not unambiguously assigned. ^13^C-NMR (100.6 MHz, CDCl_3_) δ 26.3 (2C, CH_3_), 27.9 (2C, CH_2_**C**H_2_CH_2_NH), 42.7 (2C, CH_2_N), 69.3 (2C, OCH_2_), 69.8 (2C, OCH_2_), 70.2 (2C, OCH_2_), 112.3 (2C, CH(Ph)), 118.4 (4C, CH(Ph)), 129.0 (2C, C5(Ph)), 137.7 (2C, C3(Ph)), 146.8 (2C, C1(Ph)), 198.1 (2C, CO). MS (MALDI-TOF+): Calculated for C_26_H_37_N_2_O_5_ [*M* + H] 457.2702, found 457.2672.

*((((Oxybis(ethane-2,1-diyl))bis(oxy))bis(propane-3,1-diyl))bis(azanediyl))bis(3,1-phenylene))bis(phenylmethanone)* (**32**). Obtained according to method A using trioxadiamine **1** (0.5 mmol, 110 mg), 4-iodobenzophenone (1.25 mmol, 385 mg) in the presence of CuI (19 mg) and 2-isobutyrylcyclohexanone (33 mg). Eluent CH_2_Cl_2_/MeOH 100:1 Yield 206 mg (71%). ^1^H-NMR (400 MHz, CDCl_3_) δ 1.86 (quintet, 4H, ^3^*J* = 6.0 Hz, CH_2_C**H**_2_CH_2_N), 3.27 (t, 4H, ^3^*J* = 6.4 Hz, CH_2_N), 3.56–3.60 (m, 8H, OCH_2_), 3.64–3.67 (m, 4H, OCH_2_), 5.07 (br. s, 2H, NH), 6.64 (d, 4H, ^3^*J*_obs_ = 8.8 Hz, H2, H2′ (Ph)), 7.41 (t, 2H, ^3^*J*_obs_ = 7.6 Hz, H4 (Ph′)), 7.48 (t, 4H, ^3^*J*_obs_ = 7.5 Hz, H3, H3′ (Ph′)), 7.66–7.71 (m, 8H, H3, H3′ (Ph), H2, H2′ (Ph′)). ^13^C-NMR (100.6 MHz, CDCl_3_) δ 28.3 (2C, CH_2_**C**H_2_CH_2_NH), 40.9 (2C, CH_2_N), 69.2 (2C, OCH_2_), 69.8 (2C, OCH_2_), 70.1 (2C, OCH_2_), 110.8 (4C, C2, C2′ (Ph)), 124.9 (2C, C4 (Ph′)), 127.6 (4C, CH (Ar)), 129.0 (4C, CH (Ar)), 130.7 (2C, C4 (Ph)), 132.6 (4C, CH (Ar)), 138.9 (2C, C1 (Ph′)), 152.2 (2C, C1 (Ph)), 194.7 (2C, CO). MS (MALDI-TOF+): Calculated for C_36_H_41_N_2_O_5_ [*M* + H] 581.3015, found 581.3044.

*N,N′-(((oxybis(ethane-2,1-diyl))bis(oxy))bis(propane-3,1-diyl))bis ([1,1′-biphenyl]-4-amine))* (**33**). Obtained according to method A using trioxadiamine **1** (0.5 mmol, 110 mg), 4-iodobiphenyl (1.25 mmol, 350 mg) in the presence of CuI (19 mg) and 2-isobutyrylcyclohexanone (33 mg). Eluent CH_2_Cl_2_/MeOH 200:1 Yield 170 mg (65%). ^1^H-NMR (400 MHz, CDCl_3_) δ 1.93 (quintet, 4H, ^3^*J* = 6.2 Hz, CH_2_C**H**_2_CH_2_NH), 3.29 (t, 4H, ^3^*J* = 6.6 Hz, CH_2_N), 3.62–3.66 (m, 8H, OCH_2_), 3.70–3.73 (m, 4H, OCH_2_), 3.96 (br. s, 2H, NH), 6.68 (d, 4H, ^3^*J*_obs_ = 8.6 Hz, H2, H2′ (Ph)), 7.27 (t, 2H, ^3^*J* = 7.6 Hz, H4 (Ph′)), 7.41 (t, 4H, ^3^*J*_obs_ = 7.6 Hz, H3, H3′ (Ph′)), 7.46 (d, 4H, ^3^*J*_obs_ = 8.6 Hz, H3, H3′ (Ph)), 7.66 (d, 4H, ^3^*J*_obs_ = 8.6 Hz, H2, H2′ (Ph′)). ^13^C-NMR (100.6 MHz, CDCl_3_) δ 28.7 (2C, CH_2_**C**H_2_CH_2_NH), 41.4 (2C, CH_2_N), 69.4 (2C, OCH_2_), 69.9 (2C, OCH_2_), 70.3 (2C, OCH_2_), 112.5 (4C, C2, C2′ (Ph)), 125.6 (2C, C4 (Ph′)), 125.8 (4C, CH (Ar)), 127.5 (4C, CH (Ar)), 128.3 (4C, CH (Ar)), 128.3 (2C, C4 (Ph)), 140.9 (2C, C1 (Ph′)), 147.6 (2C, C1 (Ph)). MS (MALDI-TOF+): Calculated for C_34_H_41_N_2_O_3_ [*M* + H] 525.3117, found 525.3166.

*N,N′-(((oxybis(ethane-2,1-diyl))bis(oxy))bis(propane-3,1-diyl))bis([1,1′-biphenyl]-3-amine))* (**34**). Obtained according to method B from trioxadiamine **1** (0.5 mmol, 110 mg), 3-bromobiphenyl (1 mmol, 233 mg) in the presence of Pd(dba)_2_ (2.9 mg) and BINAP (4.7 mg). Eluent CH_2_Cl_2_/MeOH 200:1 Yield 207 mg (79%). ^1^H-NMR (400 MHz, CDCl_3_) δ 1.93 (quintet, 4H, ^3^*J* = 6.2 Hz, CH_2_C**H**_2_CH_2_N), 3.29 (t, 4H, ^3^*J* = 6.6 Hz, CH_2_N), 3.62–3.66 (m, 8H, OCH_2_), 3.70–3.73 (m, 4H, OCH_2_), 4.17 (br. s, 2H, NH), 6.61 (d, 2H, ^3^*J* = 8.0 Hz, H6 (Ph)), 6.82 (s, 2H, H2 (Ph)), 6.93 (d, 2H, ^3^*J* = 7.8 Hz, H4 (Ph)), 7.25 (t, 2H, ^3^*J* = 7.3 Hz, H5 (Ph)), 7.34 (t, 4H, ^3^*J*_obs_ = 7.3 Hz, H3, H3′ (Ph′)), 7.43 (t, 2H, ^3^*J*_obs_ = 7.8 Hz, H4 (Ph′)), 7.60 (d, 4H, ^3^*J*_obs_ = 7.3 Hz, H2, H2′ (Ph′)). ^13^C-NMR (100.6 MHz, CDCl_3_) δ 28.7 (2C, CH_2_**C**H_2_CH_2_N), 41.4 (2C, CH_2_N), 69.4 (2C, OCH_2_), 70.0 (2C, OCH_2_), 70.3 (2C, OCH_2_), 111.1 (2C, CH (Ph)), 111.4 (2C, CH (Ph)), 115.8 (2C, CH (Ph)), 126.7 (2C, C4 (Ph′)), 126.8 (4C, CH (Ph′)), 128.2 (4C, CH (Ph′)), 129.2 (2C, C5 (Ph)), 141.5 (2C, C (Ar)), 141.9 (2C, C (Ar)), 148.5 (2C, C1 (Ph)). MS (MALDI-TOF+): Calculated for C_34_H_41_N_2_O_3_ [*M* + H] 525.3117, found 525.3090.

*N,N′-(((oxybis(ethane-2,1-diyl))bis(oxy))bis(propane-3,1-diyl))bis(4-methoxyaniline)* (**35**). Obtained according to method A using trioxadiamine **1** (0.5 mmol, 110 mg), 4-iodoanisole (1.25 mmol, 293 mg) in the presence of CuI (19 mg) and 2-isobutyrylcyclohexanone (33 mg). Eluent CH_2_Cl_2_/MeOH 50:1 Yield 93 mg (43%). ^1^H-NMR (400 MHz, CDCl_3_) δ 1.86 (quintet, 4H, ^3^*J* = 6.3 Hz, CH_2_C**H**_2_CH_2_N), 3.17 (t, 4H, ^3^*J* = 6.6 Hz, CH_2_N), 3.57–3.60 (m, 8H, OCH_2_), 3.65–3.68 (m, 4H, OCH_2_), 3.72 (s, 6H, OCH_3_), 6.58 (d, 4H, ^3^*J*_obs_ = 8.8 Hz, H2, H2′ (Ph)), 6.76 (d, 4H, ^3^*J*_obs_ = 8.8 Hz, H3, H3′ (Ph)), NH protons were not unambiguously assigned. ^13^C-NMR (100.6 MHz, CDCl_3_) δ 29.0 (2C, CH_2_**C**H_2_CH_2_NH), 42.9 (2C, CH_2_N), 55.7 (2C, OCH_3_), 69.7 (2C, OCH_2_), 70.1 (2C, OCH), 70.5 (2C, OCH_2_), 114.2 (4C, C2, C2′ (Ph)), 114.7 (4C, C3, C3′ (Ph)), 142.3 (2C, C1 (Ph)), 152.0 (2C, C4 (Ph)). MS (MALDI-TOF+): Calculated for C_24_H_37_N_2_O_5_ [*M* + H] 433.2702, found 433.2681.

*4-Methoxy-N-(4-methoxyphenyl)-N-(3-(2-(2-(3-((4-methoxyphenyl)amino)propoxy)ethoxy)ethoxy)propyl)aniline* (**35a**). Obtained as the second product in the synthesis of compound **35** according to method B using trioxadiamine **1** (0.5 mmol, 110 mg), 4-bromoanisole (1 mmol, 187 mg) in the presence of Pd(dba)_2_ (2.9 mg) and BINAP (4.7 mg). Eluent CH_2_Cl_2_/MeOH 100:1 Yield 11 mg (6%). ^1^H-NMR (400 MHz, CDCl_3_) δ 1.83 (quintet, 2H, ^3^*J* = 6.8 Hz, CH_2_C**H**_2_CH_2_NPh), 1.94 (quintet, 2H, ^3^*J* = 6.1 Hz, CH_2_C**H**_2_CH_2_NPh_2_), 3.26 (t, 2H, ^3^*J* = 6.6 Hz, CH_2_NPh), 3.49 (t, 2H, ^3^*J* = 6.1 Hz, CH_2_O), 3.52–3.57 (m, 2H, OCH_2_), 3.62–3.69 (m, 10H, OCH_2,_ CH_2_NPh_2_), 3.74 c (3H, OCH_3_), 3.77 с (6H, OCH_3_), 4.68 (br. s, 1H, NH), 6.78–6.82 (m, 6H, H(Ph)), 6.86–6.91 (m, 6H, H(Ph)). MS (MALDI-TOF+): Calculated for C_31_H_43_N_2_O_6_ [*M* + H] 539.312, found 539.331.

*N,N′-(((oxybis(ethane-2,1-diyl))bis(oxy))bis(propane-3,1-diyl))bis(3-methoxyaniline)* (**36**). Obtained according to method B using trioxadiamine **1** (0.5 mmol, 110 mg), 3-bromoanisole (1 mmol, 187 mg) in the presence of Pd(dba)_2_ (2.9 mg) and BINAP (4.7 mg). Eluent CH_2_Cl_2_/MeOH 100:1 Yield 158 mg (73%). ^1^H-NMR (400 MHz, CDCl_3_) δ 1.87 (quintet, 4H, ^3^*J* = 6.1 Hz, CH_2_C**H**_2_CH_2_NH), 3.21 (t, 4H, ^3^*J* = 6.5 Hz, CH_2_N), 3.57–3.62 (m, 8H, OCH_2_), 3.66–3.69 (m, 4H, OCH_2_), 3.76 c (6H, OCH_3_), 4.09 (br. s, 2H, NH), 6.15 (t, 2H, ^4^*J* = 2.2 Hz, H2 (Ph)), 6.22 (dd, 2H, ^3^*J* = 8.1 Hz, ^4^*J* = 2.2 Hz, H6 (Ph)), 6.25 (dd, 2H, ^3^*J* = 8.1 Hz, ^4^*J* = 2.2 Hz, H4 (Ph)), 7.06 (t, 2H, ^3^*J* = 8.1 Hz, H5 (Ph)). ^13^C-NMR (100.6 MHz, CDCl_3_) δ 28.9 (2C, CH_2_**C**H_2_CH_2_NH), 41.6 (2C, CH_2_N), 54.9 (2C, OCH_3_), 69.6 (2C, OCH_2_), 70.1 (2C, OCH_2_), 70.5 (2C, OCH_2_), 98.5 (2C, C2(Ph)), 101.9 (2C, C6 (Ph)), 105.8 (2C, C4 (Ph)), 129.8 (2C, C5 (Ph)), 149.9 (2C, C1 (Ph)), 160.8 (2C, C3 (Ph)). MS (MALDI-TOF+): Calculated for C_24_H_37_N_2_O_5_ [*M* + H] 433.2702, found 433.2676.

*N-(3-(2-(2-(3-aminopropoxy)ethoxy)ethoxy)propyl)-3-methoxyaniline* (**36a**). Obtained as the second product in the synthesis of compound **36** according to method A using trioxadiamine **1** (0.5 mmol, 110 mg), 3-iodoanisole (1.25 mmol, 293 mg) in the presence of CuI (19 mg) and 2-isobutyrylcyclohexanone (33 mg). Eluent CH_2_Cl_2_/MeOH 20:1 Yield 28 mg (17%). ^1^H-NMR (400 MHz, CDCl_3_) δ 1.76 (quintet, 2H, ^3^*J* = 6.0 Hz, CH_2_C**H**_2_CH_2_NH_2_), 1.87 (quintet, 2H, ^3^*J* = 6.2 Hz, CH_2_C**H**_2_CH_2_NPh), 3.19 (t, 2H, ^3^*J* = 6.5 Hz, C**H**_2_NH_2_), 3.39 (q, 2H, ^3^*J* = 6.2 Hz, CH_2_N), 3.56–3.59 (m, 8H, OCH_2_), 3.66–3.68 (m, 4H, OCH_2_), 3.76 c (3H, OCH_3_), 5.28 (br. s, 1H, NH), 6.18 (t, 1H, ^4^*J* = 2.2 Hz, H2 (Ph)), 6.23–6.26 (m, 2H, H4, H6 (Ph)), 7.05 (t, 2H, ^3^*J* = 8.0 Hz, H5 (Ph)). NH_2_ were not unambiguously assigned. ^13^C-NMR (100.6 MHz, CDCl_3_) δ 28.4 (1C, OCH_2_**C**H_2_CH_2_N), 28.6 (1C, OCH_2_**C**H_2_CH_2_N), 36.6 (1C, CH_2_NH_2_), 41.3 (1C, CH_2_NHPh), 55.0 (1C, OCH_3_), 69.7 (1C, OCH_2_), 70.0 (2C, OCH_2_), 70.1 (2C, OCH_2_), 70.4 (1C, OCH_2_), 99.5 (1C, C2(Ph)), 103.0 (1C, C6 (Ph)), 106.7 (1C, C4 (Ph)), 129.9 (1C, C5 (Ph)), 148.4 (1C, C1 (Ph)), 161.4 (1C, C3 (Ph)). MS (MALDI-TOF+): Calculated for C_17_H_31_N_2_O_4_ [*M* + H] 327.2284, found 327.2261.

*(((((Oxybis(ethane-2,1-diyl))bis(oxy))bis(propane-3,1-diyl))bis(azanediyl))bis(3,1-phenylene))bis(phenylmethanone)* (**37**). Obtained according to method B using trioxadiamine **1** (0.5 mmol, 110 mg), 3-bromobenzophenone (1 mmol, 187 mg) in the presence of Pd(dba)_2_ (2.9 mg) and BINAP (4.7 mg). Eluent CH_2_Cl_2_/MeOH 200:1 Yield 238 mg (82%). ^1^H-NMR (400 MHz, CDCl_3_) δ 1.89 (quintet, 4H, ^3^*J* = 6.1 Hz, CH_2_C**H**_2_CH_2_N), 3.25 (t, 4H, ^3^*J* = 6.4 Hz, CH_2_N), 3.57–3.60 (m, 8H, OCH_2_), 3.63–3.66 (m, 4H, OCH_2_), 6.81 (dd, 2H, ^3^*J* = 8.0 Hz, ^4^*J* = 2.2 Hz, H6 (Ph)), 7.00–7.04 (m, 4H, H4, H6 (Ph)), 7.21 (t, 2H, ^3^*J*_obs_ = 7.8 Hz, H5(Ph)), 7.44 (t, 4H, ^3^*J*_obs_ = 7.8 Hz, H3, H3′ (Ph′))), 7.65 (t, 2H, ^3^*J* = 7.8 Hz, H4 (Ph′)), 7.88 (d, 4H, ^3^*J*_obs_ = 7.1 Hz, H2, H2′ (Ph′)). ^13^C-NMR (100.6 MHz, CDCl_3_) δ 28.3 (2C, OCH_2_**C**H_2_CH_2_NH), 41.7 (2C, CH_2_N), 69.4 (2C, OCH_2_), 69.8 (2C, OCH_2_), 70.2 (2C, OCH_2_), 113.1 (2C, C6 (Ph)), 116.6 (2C, C4 (Ph)), 119.1 (2C, C2 (Ph)), 127.7 (4C, C3, C3′ (Ph′)), 128.4 (2C, C5 (Ph)), 129.6 (4C, C2, C2′ (Ph′)), 131.8 (2C, C4 (Ph′)), 137.5 (2C, C1 (Ph′) or C3 (Ph)), 138.1 (2C, C3 (Ph) or C1 (Ph′)), 147.9 (2C, C1 (Ph)), 196.8 (2C, CO). MS (MALDI-TOF+): Calculated for C_36_H_41_N_2_O_5_ [*M* + H] 581.3015, found 581.3070.

## 4. Conclusions

In summary, the following regularities can be ruled out from the comparison of copper- and palladium-catalyzed diarylation of oxadiamines. Most of peculiarities are evoked by the fluorine-containing halogenobenzenes. The copper-catalyzed arylation with fluoroiodobenzenes proceeds better with 3-fluoroiodobenzene, and the results are comparable with those obtained with palladium catalysis. To synthesize 4- and 2-fluorophenyl derivatives, palladium-catalyzed amination is preferable, although in certain cases, copper catalysis brings enough good results. Even the Cu(I)-catalyzed amination of 4-bromofluorobenzene was shown to be possible. Similarly, the Cu(I)-mediated arylation with 3-iodo(trifluoromethyl)benzene provides excellent yields of the corresponding diaryl derivatives of oxadiamines, and the synthesis of compounds with *para*- or *ortho*-(trifluoromethyl)phenyl groups should be accomplished via Pd(0)-catalyzed amination. As for the other halogenobenzenes, copper- and palladium-catalyzed aminations produce similar results in the *N,N′*-diarylation of oxadiamines; moreover, acetyl-containing derivatives can be successfully obtained only in the Cu(I)-catalyzed reactions.

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
