# Peer review of "Cu(I)- and Pd(0)-Catalyzed Arylation of Oxadiamines with Fluorinated Halogenobenzenes: Comparison of Efficiency"

_molecules, 2020, doi:10.3390/molecules25051084_

Round 1
Reviewer 1 Report
The manuscript by Beletskaya, Averin with coworkers deals with an important topic, that is the application of copper-catalyzed amination reactions to the synthesis of N,N’-diaryl derivatives of oxadiamines, and the comparison of this approach with a more familiar palladium-catalyzed amination. The target molecules are of interest as they possess fluorine-containing substituents like fluorophenyl, (trifluoromethyl)phenyl groups, moreover, N,N’-di(hetero)aryl substituted oxadiamines can be regarded as potential physiologically active compounds bearing two identical pharmacophore groups combined with a linker which structure can be adjusted. The manuscript is written in a good manner, it provides the background for the research, the organization of the material is quite appropriate, authors carried out enough thorough investigations as they tried various fluorine-containing aryl halides, different oxadiamines, they tested the possibility to introduce aryl bromides in the copper-catalyzed reactions. Also where possible they made comparisons with the alternative Pd-catalyzed reactions and found out the processes where copper-mediated reactions could be preferable affording higher yields. All compounds obtained are adequately characterized and experimental procedures are clearly described. I only recommend the addition of some more references to the Introduction section, e.g. doi.10.1070/RCR4871 and doi.10.1021/cr500465n. Thus I support the publication of this manuscript.
Author Response
The proposed references have been inclided in the Introduction section.
Reviewer 2 Report
In this paper, the selective synthesis of N,N’-disubstituted (diarylated) trioxadiamines has been described. The authors demonstrated the Pd- and Cu-catalyzed cross-coupling involving C-N bond formation. These findings in this paper are of basic interest and the products are useful for some fields of chemistry, I think. Therefore, I recommend publication in Molecules after modification.
Note:
1) Page 4, line 115; A universal catalytic system Pd(dba)2/rac-BINAP -> Pd(dba)2/rac-BINAP …, one of the efficient catalyst systems for amination,
Author Response
The proposed modification of the sentence has been done.
Reviewer 3 Report
The paper submitted by Averin et al. present a new strategy for the arylation of oxadiamines with fluorinated halogenobenzenes under cu and pd catalysis. versa the paper is well presented and could be accepted for publication in Molecules after the following revision.
the referee suggest to add some general review on copper catalysis, for instance Evaon Book, Taillefer ACIE 2009...
Author Response
Additional references of general character have been included in the Introduction section (refs. 13-17).
Reviewer 4 Report
In previous papers, the Buchwald-Hartwig arylation of amines were investigated by the authors. In a particular case, a successful diarylation of diamines were accomplished with the aid of copper catalysts, thus substituting the expensive palladium complexes.
In the present paper, the same chemistry was extended to oxadiamines, interesting building blocks included into some bioactive compounds. The synthetic problem that was investigated in the present paper was connected to the presence of the fluorine atom in the haloaryl compounds that should be aminated.
Different copper and palladium catalysts were compared in the amination of fluorinated halobenzenes with a series of oxadiamines and an impressively large number of compounds were synthesised and characterised.
Taking into account these facts, the paper can be accepted in Molecules, but I believe that a revision should be undertaken to improve the article.
The main defect that I see is that the paper is difficult to follow, due to many unsatisfactory results that are present in the Tables, together with a not optimal presentation of the results. For example, in Table 1 the copper catalysed undesired products of mono-amination are shown together with those of bis-alkylation, and the same is done in Table 2 (palladium catalysed reaction). Probably, the Tables 1 and 2 will result clearer if copper and palladium catalysed reactions will be grouped together into a single Table to yield the bis-amination products, and mono-amination products will be shown later. Thus, a reader can evaluate in which circumstances a copper catalysis can substitute the precious metal process, comparing the same reactions that lead to the same products.
A similar approach can be extended also to Table 3 and 4, and 5 and 6.
Moreover, unsatisfactory results should be reported only when they are really useful for the discussion. Otherwise, they can be summarised and reported in more details into the Supporting Information Section.
In this respect, in my opinion, the 2.4 paragraph (Formation of amino alcohols and diols as a side reaction, p. 9) is still in a preliminary state (according to the text “It is too prematurely to propose a mechanism”), and thus I believe that this paragraph can be summarised in a few lines and more details should be moved to the Supporting Information Section.
Minor points.
There are some mistakes/misprints that can be easily corrected, such as “flurobenzene”, line 112;“formt”, line 144; “tetraryalted”, line 152; “th”, line 224; “ananlogous”, line 230; “prematurely”, line 235; “anilne”, line 302, line 323, line 344; “obatined”, line 811; “methanone”, line 747 and line 825, and so on.
Author Response
The main defect that I see is that the paper is difficult to follow, due to many unsatisfactory results that are present in the Tables, together with a not optimal presentation of the results. For example, in Table 1 the copper catalysed undesired products of mono-amination are shown together with those of bis-alkylation, and the same is done in Table 2 (palladium catalysed reaction). Probably, the Tables 1 and 2 will result clearer if copper and palladium catalysed reactions will be grouped together into a single Table to yield the bis-amination products, and mono-amination products will be shown later. Thus, a reader can evaluate in which circumstances a copper catalysis can substitute the precious metal process, comparing the same reactions that lead to the same products. A similar approach can be extended also to Table 3 and 4, and 5 and 6.
This proposal has been taken into account when preparing a corrected version of the manuscript. Tables 1 and 2, 3 and 4, 5 and 6 have been merged, insignificant results have been withdrawn. The text in proper places has been modified according to the changes made in Tables. the changes are marked in green.
Moreover, unsatisfactory results should be reported only when they are really useful for the discussion. Otherwise, they can be summarised and reported in more details into the Supporting Information Section.
Some results of low importance have been moved to Supporting Information file.
In this respect, in my opinion, the 2.4 paragraph (Formation of amino alcohols and diols as a side reaction, p. 9) is still in a preliminary state (according to the text “It is too prematurely to propose a mechanism”), and thus I believe that this paragraph can be summarised in a few lines and more details should be moved to the Supporting Information Section.
This has been also done and the Scheme together with experimental details of these compounds have been transferred to Supporting Information file.
Minor points.
There are some mistakes/misprints that can be easily corrected, such as “flurobenzene”, line 112;“formt”, line 144; “tetrarylated”, line 152; “th”, line 224; “ananlogous”, line 230; “prematurely”, line 235; “anilne”, line 302, line 323, line 344; “obatined”, line 811
The text has been checked and misprints have been corrected.
“methanone”, line 747 and line 825 - .
“methanone” is not a misprint because it is the part of the name designating the ketone function.